# Ethanol exposure increases mutation rate through error-prone polymerases

Karin Voordeckers[1,2], Camilla Colding[3], Lavinia Grasso[4], Benjamin Pardo [4], Lore Hoes[1,2,5,6],
Jacek Kominek [1,2,11], Kim Gielens[1,2], Kaat Dekoster [1,2], Jonathan Gordon[1,2], Elisa Van der Zande [1,2],
Peter Bircham[1,2], Toon Swings[7,8], Jan Michiels [7,8], Peter Van Loo [9,10], Sandra Nuyts[5,6], Philippe Pasero [4],
Michael Lisby [3✉] & Kevin J. Verstrepen [1,2✉]

Ethanol is a ubiquitous environmental stressor that is toxic to all lifeforms. Here, we use the model eukaryote *Saccharomyces cerevisiae* to show that exposure to sublethal ethanol concentrations causes DNA replication stress and an increased mutation rate. Specifically, we find that ethanol slows down replication and affects localization of Mrc1, a conserved protein that helps stabilize the replisome. In addition, ethanol exposure also results in the recruitment of error-prone DNA polymerases to the replication fork. Interestingly, preventing this recruitment through mutagenesis of the PCNA/Pol30 polymerase clamp or deleting specific error-prone polymerases abolishes the mutagenic effect of ethanol. Taken together, this suggests that the mutagenic effect depends on a complex mechanism, where dysfunctional replication forks lead to recruitment of error-prone polymerases. Apart from providing a general mechanistic framework for the mutagenic effect of ethanol, our findings may also provide a route to better understand and prevent ethanol-associated carcinogenesis in higher eukaryotes.

[1] Laboratory of Systems Biology, VIB-KU Leuven Center for Microbiology, Leuven, Belgium. [2] Laboratory for Genetics and Genomics, Center of Microbial and Plant Genetics, Department M2S, KU Leuven, Gaston Geenslaan 1, 3001 Heverlee, Belgium. [3] Department of Biology, University of Copenhagen, Ole Maaloees Vej 5, DK-2200 Copenhagen N, Denmark. [4] Institut de Génétique Humaine, CNRS and Université de Montpellier, 141 rue de la Cardonille, Montpellier, France. [5] Laboratory of Experimental Radiotherapy, Department of Oncology, KU Leuven, UZ Herestraat 49, 3000 Leuven, Belgium. [6] Department of Radiation Oncology, Leuven Cancer Institute, UZ Leuven, 3000 Leuven, Belgium. [7] Centre of Microbial and Plant Genetics, KU Leuven, Kasteelpark Arenberg 20, 3001 Leuven, Belgium. [8] VIB-KU Leuven Center for Microbiology, Kasteelpark Arenberg 20, 3001 Leuven, Belgium. [9] The Francis Crick Institute, 1 Midland Road, London, UK. [10] Department of Human Genetics, KU Leuven, Leuven, Belgium. [11] Present address: Laboratory of Genetics, Genome Center of Wisconsin, J. F. Crow Institute for the Study of Evolution, Wisconsin Energy Institute, University of Wisconsin-Madison, Madison, WI 53706, USA. ✉email: mlisby@bio.ku.dk; Kevin.Verstrepen@kuleuven.vib.be

Ethanol is a ubiquitous natural compound produced as a primary metabolite by several yeasts and bacteria. In high concentrations, ethanol is toxic to all lifeforms. Several large-scale studies in model systems like *Saccharomyces cerevisiae* and *Escherichia coli* reveal multiple, complex targets of ethanol, including cellular membranes, protein stability, telomere length homeostasis, and cell cycle control[1–4]. Apart from this short-term toxicity, prolonged excessive ethanol intake is associated with multiple diseases and a decreased life expectancy in humans[5]. Epidemiological studies indicate a strong correlation between alcohol intake and the risk of developing specific types of cancers[6,7]. Most tumors form at sites where tissues come into direct contact with ethanol, such as the mouth, upper throat, and esophagus[6–8].

Despite the clear link between ethanol intake and the incidence of specific tumors, the exact molecular mechanisms underlying the carcinogenic effect of ethanol are still not fully understood. Interestingly, the potential mutagenic effect of ethanol has also not been extensively researched in other (model) systems. It is known that several stressors, such as nutrient starvation, drug treatment, and high salinity can affect mutation rates and genome stability across multiple organisms[9]. The best-studied system is arguably that of stress-induced mutagenesis (SIM) in bacteria[10]. Multiple bacterial species display increased mutation rates or altered mutational spectra when exposed to stressors, such as low doses of antibiotics or nutritional stresses[11–13]. Although ionizing radiation or alkylating agents can directly modify DNA bases, other stressors such as proteotoxic stress do not directly cause DNA damage but can trigger mutagenic stress responses. SIM encompasses multiple signaling pathways, including the SOS DNA damage response, the RpoS general stress response, and the RpoE membrane protein stress response[14–16]. In many cases, DNA polymerases with a lower replication fidelity, the so-called translesion polymerases or error-prone polymerases, play a central role[15,16]. These error-prone polymerases are induced or recruited upon stress. As they replace the higher-fidelity replicative polymerases, more mutations are introduced when DNA is synthesized. Interestingly, a recent study showed that alcohol-associated cancers display error-prone polymerase-associated mutational spectra, although the exact mechanism by which these polymerases are involved and/or are affected by ethanol remained unclear[17].

Environmental stress can also affect genome stability in eukaryotes. The pathogenic yeast *Candida albicans* displays gross chromosomal rearrangements and aneuploidies when treated with fluconazole[18]. Different types of stress have been reported to alter chromosome segregation and mutation rate in *S. cerevisiae*[19,20]. For example, stresses that cause protein misfolding are associated with increased mutation rates and aneuploidy[19,21]. SIM has also been reported in multicellular eukaryotes, although the exact mechanistic details are often not yet known. In *Arabidopsis thaliana*, e.g., high salinity leads to accumulation of a distinct set of mutations[22]. In mammalian cells, hypoxic environments result in increased mutation rates and genome instability by suppressing error-free DNA repair pathways[23,24], and osmotic stress causes DNA damage and an increased mutation rate[25]. More generally, defective replisomes and replication stress also cause genome instability and ultimately cancer in higher eukaryotes[26–28]. Regions of single-stranded DNA (ssDNA), formed at dysfunctional and stalled replication forks or at DNA double-strand breaks (DSB), can underlie local, transient hypermutability in both yeast cells and malignant tumors[29].

In this study, we report that sublethal, naturally occurring levels of ethanol cause proteotoxic and replication stress in the model eukaryote *S. cerevisiae*. We find that ethanol increases mutation rates. Our results show that ethanol exposure slows down DNA replication and affects localization of Mrc1, a highly conserved component of the replisome required for efficient replication[30]. Moreover, this ethanol-associated genetic instability relies on the recruitment of error-prone polymerases to the replication fork. Together, our results shed light on the mechanisms underlying ethanol-related genome instability. Our findings reveal the framework of a complex chain of events when yeast cells are exposed to ethanol and also serve as a starting point to better understand how ethanol increases mutagenesis in higher eukaryotes.

## Results

**Ethanol is mutagenic.** To better understand how cells react and adapt to increasing levels of ethanol, we previously performed a long-term evolution experiment[3], where non-ethanol tolerant yeast cells were exposed to gradually increasing ethanol levels[3]. Notably, the average number of single-nucleotide polymorphisms in evolved clones, as determined by whole-genome sequencing, was higher than expected, based on reported spontaneous mutation rates as well as compared with other long-term evolution experiments[31,32]. This led us to hypothesize that ethanol exposure caused an increased mutation rate.

To investigate the effect of ethanol on eukaryotic genome stability, we determined mutation rates in *S. cerevisiae* cells exposed to different ethanol levels, using the *CAN1* gene as a mutation reporter in a series of fluctuation assays[33,34]. *Can1⁻* cells can grow on medium containing canavanine, a toxic arginine analog. Determining the number of *can1⁻* (canavanine resistant, *can^R*) mutants in populations exposed to ethanol allows calculating the mutation rate. This well-established reporter assay demonstrated that mutation rates increase with increasing ethanol concentrations (Fig. 1a). It should be noted that the ethanol concentrations used in these experiments are relatively low for yeast and did not reduce cell viability (Fig. 1c). Importantly, in each of the experiments below, mutation rates are always compared within one experiment. Although there is experiment-to-experiment variation in the absolute mutation rate (as is always the case when performing fluctuation assays), the fold changes in mutation rate were always comparable—ranging from 2.6- to 3.5-fold increase when comparing 6% ethanol to 0% ethanol conditions for S288c wild-type (WT) strain.

We next tested the effect of ethanol on mutation rates in RM11-1a, a haploid derivative of a feral vineyard isolate. This strain is phenotypically and genotypically distinct from the lab strain S288c[35]. Ethanol also increases the mutation rate in this strain (Fig. 1b). The mutagenic effect was also observed in fluctuation assays employing a different mutation reporter, *URA3*, where mutants are selected on 5-Fluoro-orotic Acid (FOA) (Fig. 1d and "Methods"). In agreement with previous reports[34], we found that some FOA^R colonies did not carry mutations in *URA3*. All subsequent mutation reporter assays were therefore performed using a canavanine-based fluctuation assay. W303, another commonly used lab strain, is notoriously less ethanol tolerant than S288c because of a mutation in *SSD1*, a gene involved in cell wall integrity signaling[36]. Introducing the S288c *SSD1* allele into W303 increases its ethanol tolerance, but the strain still grows poorly in ethanol compared with S288c. Determining mutation rates in such a strain using a standard fluctuation assay approach proved difficult, because the cells did not grow well when exposed to ethanol over longer timespans. However, we did find that even a short exposure to ethanol causes an increase in *can^R* mutant frequency in W303 (Supplementary Fig. 1).

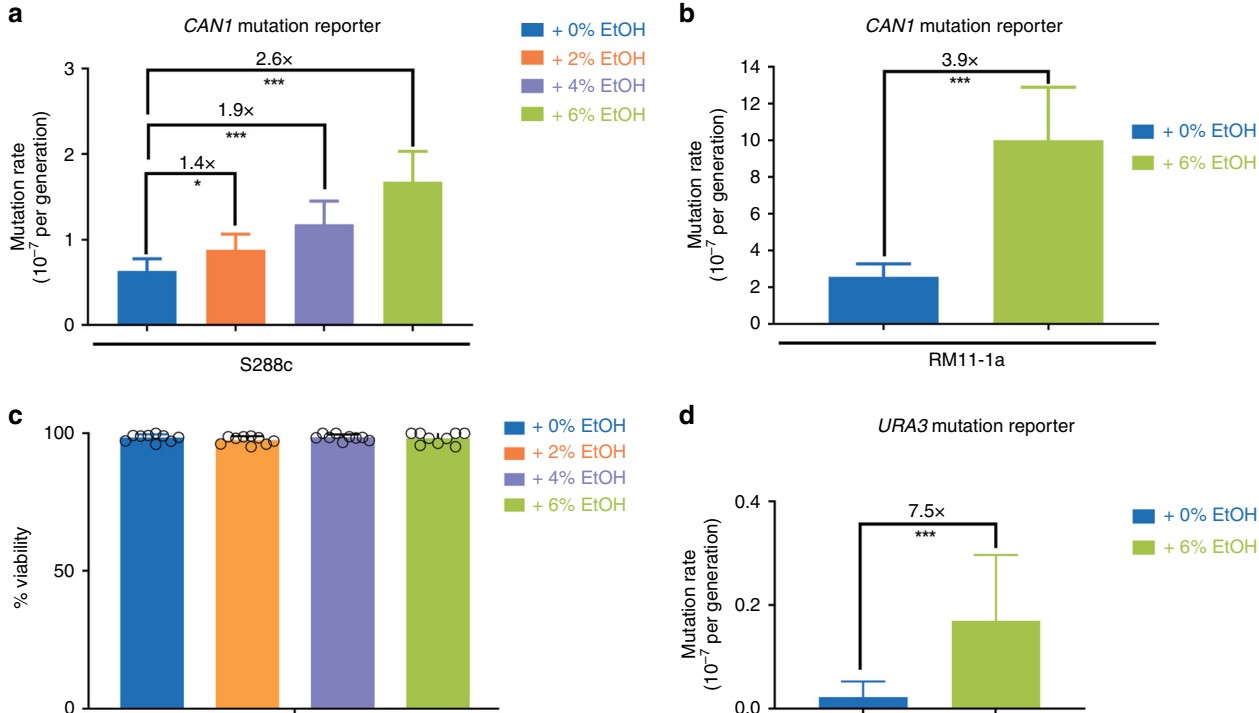

**Fig. 1 Ethanol increases mutation rate. a, b** Mutation rate increases with ethanol concentrations. Cultures of *S. cerevisiae* strain S288c (strain VK111) (**a**) and RM11-1a (**b**) were grown in synthetic media (2% glucose) and indicated ethanol concentrations (v/v). For each condition, 54 cultures were analyzed. Data represent mutation rate estimates, as determined by fluctuation assays on canavanine, error bars represent 95% confidence intervals. For more details, see "Methods" section. Statistical significance of differences in mutation rates was assessed using a likelihood ratio test. *$P < 0.05$, ***$P < 0.001$. Specifically, for S288c, p-values are as follows: 0–2%: $p = 0.0290$; 0–4%: $p = 1.035 \times 10^{-4}$, 0–6%: $p = 2.962 \times 10^{-10}$. For RM11-1a, $p = 2.756 \times 10^{-4}$. **c** Ethanol does not affect cell viability. Cells of strain VK111 were grown in synthetic media (2% glucose), at different ethanol concentrations (v/v) for the same time as a standard fluctuation assay. Cell viability was determined using methylene blue staining. Bars represent average of nine biological replicate measurements ± SD. Error bars are clipped at 100%. At least 618 cells were analyzed per ethanol concentration. **d** Effect of ethanol on mutation rate is also observed using *URA3* mutation reporter. Cells of strain VK111 were grown in synthetic media (2% glucose) at different ethanol concentrations (v/v). For each condition, 54 cultures were analyzed. Mutation rate estimates, as determined by fluctuation assays on 5-fluoro-orotic acid (FOA), are shown. Error bars represent 95% confidence intervals. Statistical significance of differences in mutation rates was assessed using a likelihood ratio test. ***$P < 0.001$; more specifically, $p = 2.756 \times 10^{-4}$. Source data for this figure are provided as a Source Data file.

Taken together, these data indicate that the observed ethanol-associated increase in mutation rate is independent of genetic background and reporter assay used.

**Mutagenic effect of ethanol depends on acetaldehyde.** The carcinogenic effects of ethanol in mammalian cells have been mostly considered to be caused by metabolism of ethanol to acetaldehyde. Acetaldehyde can form mutagenic and carcinogenic DNA adducts and cause interstrand crosslinks and DSBs, both in vivo and in vitro[37]. Genetic linkage studies have shown that individuals with mutations in acetaldehyde-metabolizing enzymes display an increased risk for tumors of the upper gastrointestinal tract[38]. However, such individuals display excessively high acetaldehyde levels compared with individuals carrying WT alleles[39]. Hence, it is still unclear whether and to what extent physiological acetaldehyde contributes to the carcinogenic effects of ethanol, and whether other molecular mechanisms are also involved.

To investigate the importance of acetaldehyde in mediating the observed ethanol-induced mutation rate increase in *S. cerevisiae*, we used a combination of genetic and chemical approaches. We first tested whether extracellular addition of acetaldehyde would increase mutation rate. Cells exposed to high (>0.1% v/v) acetaldehyde levels did not grow, corroborating the toxic effects

of acetaldehyde. Surprisingly, none of the sublethal acetaldehyde levels tested resulted in an increased mutation rate (Fig. 2a).

As acetaldehyde is extremely volatile, the lack of detectable mutagenic effect could be due to acetaldehyde evaporation during the experiment. To minimize evaporation, we next added acetaldehyde using a pre-chilled syringe to a chilled and sealed glass vial containing yeast cells. As this low temperature did not allow for cell growth, we could only determine the number of $can^R$ mutants after acetaldehyde exposure and not an absolute mutation rate (i.e., the number of mutations per generation). In line with our previous findings, we did not observe an increase in $can^R$ mutant frequency after acetaldehyde exposure (Supplementary Fig. 2a).

The *S. cerevisiae* genome encodes 5 alcohol dehydrogenase genes (*ADH1–5*) involved in ethanol metabolism, with *ADH2* encoding the enzyme responsible for converting ethanol to acetaldehyde[40]. Deleting *ADH2* did not affect the mutagenic effect of ethanol and overexpressing *ADH2* did not further increase mutation rate upon ethanol exposure (Fig. 2b). These results would indicate that acetaldehyde is not required for the mutagenic effect of ethanol. However, as acetaldehyde levels did not appear to be altered in these strains, as determined using two different approaches (high-performance liquid chromatography (HPLC) and enzymatic), we cannot draw any definite conclusions about the contribution of acetaldehyde to ethanol-induced

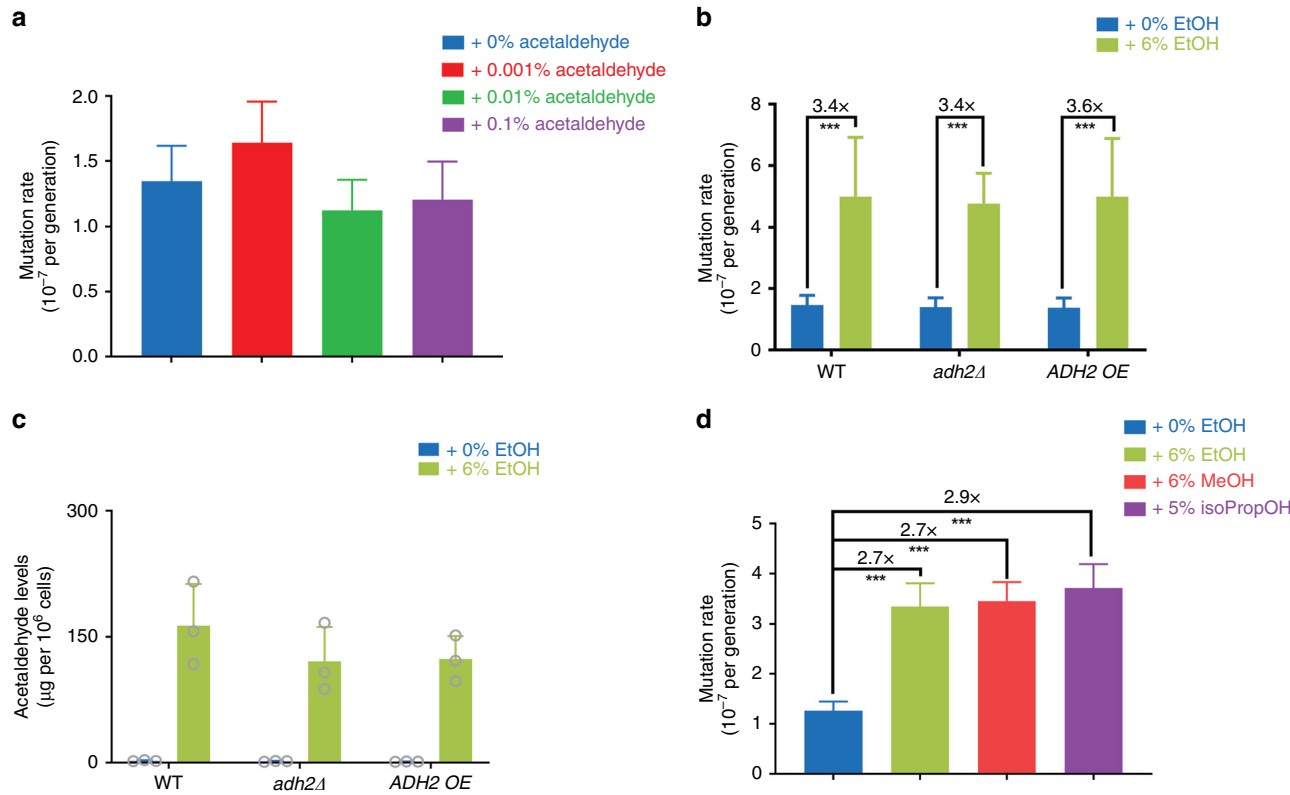

**Fig. 2 Role of acetaldehyde in mutagenic effect of ethanol. a** Extracellular addition of acetaldehyde does not alter mutation rate. Cells (VK111) were grown in synthetic media (2% glucose), supplemented with the indicated acetaldehyde concentrations (v/v). For each condition, 54 cultures were analyzed. Data represent mutation rate estimates, as determined by fluctuation assays on canavanine. Error bars represent 95% confidence intervals. Statistical significance of differences in mutation rates was assessed using a likelihood ratio test. **b** Altered *ADH2* levels do not abolish the mutagenic effect of ethanol. Cells (VK111, EV14, and EV19) were grown in synthetic media (2% glucose) at different ethanol concentrations (v/v). For MVP11 in 6% ethanol, 36 cultures were analyzed. For all other conditions, 54 cultures were analyzed. Data represent mutation rate estimates, as determined by fluctuation assays on canavanine. Error bars represent 95% confidence intervals. Statistical significance of differences in mutation rates was assessed using a likelihood ratio test. ***$P < 0.001$. Specifically, $p$-values are $2.204 \times 10^{-10}$, $4.108 \times 10^{-15}$, and 0 for WT, *adh2Δ*, and *ADH2* OE, respectively. **c** Acetaldehyde levels are not altered in *adh2* mutant cells. Strains VK111, EV14, and EV19 were grown in synthetic media (2% glucose) with 0 or 6% ethanol added. Bars represent average of three biological replicates per strain ± SD. Acetaldehyde levels were determined enzymatically using a Megazyme Acetaldehyde Assay.
**d** Methanol and isopropanol also increase mutation rate. Cells (VK111) were grown in synthetic media (2% glucose) at different alcohol concentrations (v/v). EtOH ethanol, MeOH methanol, isoPropOH isopropanol. For each condition, 108 cultures were analyzed. Data represents mutation rate estimates, as determined by fluctuation assays on canavanine. Error bars represent 95% confidence intervals. Statistical significance of differences in mutation rates was assessed using a likelihood ratio test. ***$P < 0.001$. $P$-values for all comparisons were reported as 0. Source data for this figure are provided as a Source Data file.

mutagenesis based on these experiments (Fig. 2c and Supplementary Fig. 2b).

To get more insight into a potential role of acetaldehyde in the mutagenic effect of ethanol, we next used fomepizole, a well-established alcohol dehydrogenase inhibitor[41,42]. Fomepizole addition did not abolish the mutagenic effect of ethanol, but also did not appear to change acetaldehyde levels, again preventing us from drawing any definite conclusions on the contribution of acetaldehyde to the mutagenic effect of ethanol (Supplementary Fig. 2c, d).

Acetaldehyde is reported to cause DNA interstrand crosslinks and DSBs[43]. Non-homologous end-joining (NHEJ) helps maintain genome integrity by aiding repair of aldehyde-induced DSBs[44]. Deleting *YKU70*, a key NHEJ component, does not further increase mutation rate in ethanol-exposed cells (Supplementary Fig. 3a), suggesting that the mutagenic effect of ethanol might not be mediated through acetaldehyde-induced DSBs. Acetaldehyde can also form DNA adducts and some of these adducts form DNA interstrand crosslinks[7,43,45]. *PSO2* encodes a nuclease involved in repair of DNA breaks that result from

interstrand crosslinks and strains deficient in *PSO2* show low mutability in response to interstrand crosslink-inducing mutagens[46]. Deletion of *PSO2* does not abolish the mutagenic effect of ethanol (Supplementary Fig. 3b), suggesting that acetaldehyde might not be the principal mediator of the mutagenic effect of ethanol, or at least that the underlying mechanism may be more complex than direct chemical modification of DNA by acetaldehyde.

In another effort to investigate if the mutagenic effect of ethanol depends on its metabolism to acetaldehyde, we tested the effect of other alcohols. Methanol and isopropanol both cause an increase in mutation rate (Fig. 2d). Methanol is metabolized by yeast cells to formaldehyde, a known mutagen. Isopropanol on the other hand cannot be metabolized by yeast cells. These data indicate that alcohol metabolism is not required to increase mutation rate.

We next performed additional fluctuation assays using a total of five genetically different yeast strains, four acetaldehyde concentrations, and two mutation reporters (*URA3* and *CAN1*). Our results show that in some, but not all of the yeast strains

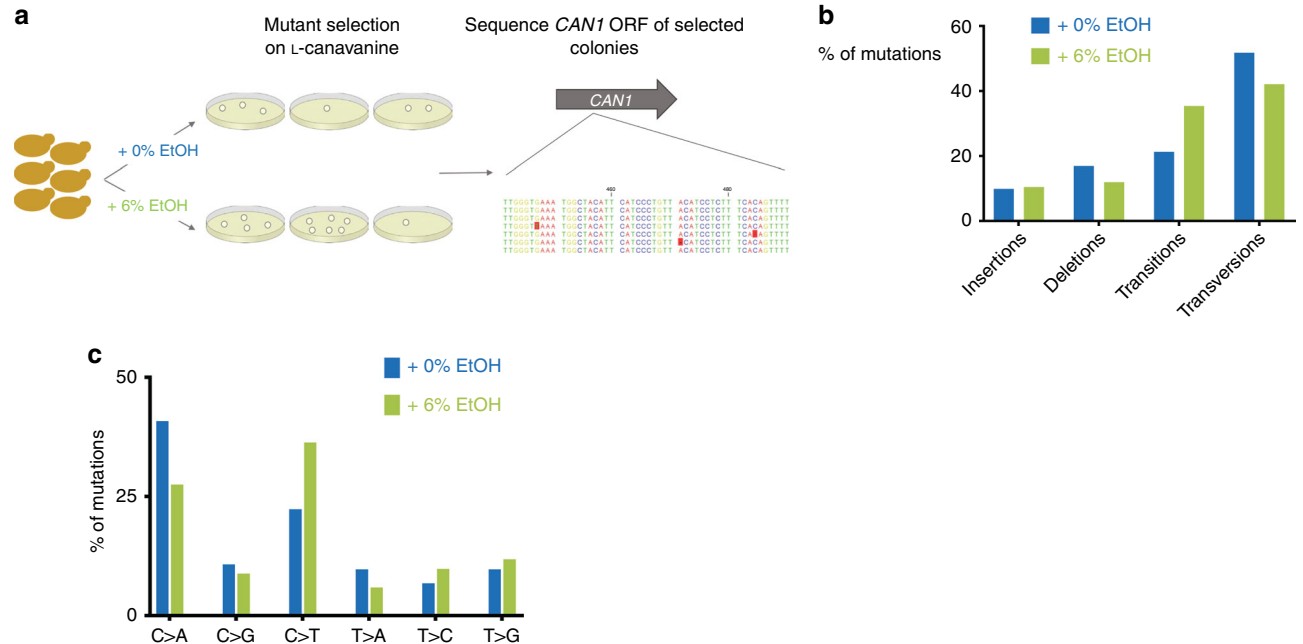

**Fig. 3 Mutation spectra of $can^R$ colonies in 0 and 6% ethanol. a** Experimental setup. Cells were grown in medium supplemented with the indicated ethanol concentrations. $can^R$ colonies were identified by plating on medium containing canavanine. The *CAN1* ORF of these colonies was sequenced to identify mutations. For 0% EtOH, a total of 121 colonies were analyzed; for 6% EtOH, a total of 113 colonies were analyzed. **b** Overall distribution of mutations identified in $can^R$ colonies isolated after exposure to 0 or 6% ethanol. For 0% EtOH, a total of 121 colonies were analyzed; for 6% EtOH, a total of 113 colonies were analyzed. **c** Base-pair substitution spectra of $can^R$ colonies in 0 and 6% ethanol. For 0% EtOH, a total of 121 colonies were analyzed; for 6% EtOH, a total of 113 colonies were analyzed. Source data for this figure are provided as a Source Data file.

tested, acetaldehyde increases mutation rate (Supplementary Fig. 4). We also noticed that the highest acetaldehyde concentration used in our assays (0.1%) was borderline lethal, suggesting that acetaldehyde clearly affected the cells. These results indicate that the lack of mutagenic effect of acetaldehyde observed in some strains is not due to a technical problem and provide additional support that the mutagenic effect of ethanol observed in yeast is not solely due to acetaldehyde.

Taken together, our results do not support the central role of acetaldehyde in the mutagenic effect of ethanol, although it has to be noted that manipulating and measuring acetaldehyde concentrations is difficult.

**Ethanol-exposed cells do not accumulate ROS.** The carcinogenic effects of ethanol have also been linked to reactive oxygen species (ROS) produced during ethanol metabolism[7]. ROS can cause lipid peroxidation and the subsequent formation of (mutagenic) DNA adducts. We assessed ROS production in ethanol-treated cells using the cell permeant reagent $H_2$DCFDA (2',7'-dichlorofluoroscein diacetate), a commonly used oxidant-sensitive probe[47]. After diffusion into the cell, $H_2$DCFDA is first deacetylated by cellular esterases. In the presence of ROS, this probe is then readily oxidized into a fluorescent compound. As expected, exposing cells to hydrogen peroxide increased oxidant levels ($p = 0.0029$, unpaired $t$-test with Welch's correction) (Supplementary Fig. 5). Ethanol-exposed cells on the other hand do not show an increase in fluorescence. These results indicate that ethanol exposure does not cause an increase in ROS.

**Mutations indicate that ethanol affects replication.** If the mutagenic effect of ethanol is caused by specific mutational mechanisms, this could result in a specific mutational spectrum. To investigate this, we sequenced the *CAN1* ORF of canavanine-resistant colonies, isolated from fluctuation assays performed in 0

and 6% ethanol conditions. A total of 234 sequences were analyzed (121 for 0% and 113 for the 6% (v/v) ethanol condition; see also "Methods," Fig. 3a, and Supplementary Data 1). All sequenced colonies contained at least one mutation in the *CAN1* ORF. We did not find a significant difference between the location and distribution of mutations in 0 or 6% ethanol conditions (Supplementary Fig. 6).

We generally found more transitions in cells that had been exposed to 6% ethanol, although the overall distribution of broadly defined mutation classes (insertions and deletions, transitions, and transversions) did not appear to be markedly different between 0 and 6% ethanol (Fig. 3b, $\chi^2$-test, $p = 0.0611$).

Interestingly, both in vivo and in vitro studies have indicated that acetaldehyde causes GG-to-TT mutations, due to the formation of GG intrastrand crosslinks[45,48]. The *CAN1* ORF contains 180 GG pairs and some of these can generate stop codons and non-synonymous mutations when mutated to TT. Hence, it seems plausible that some GG-to-TT mutations would generate canavanine-resistant colonies. No GG-to-TT mutations were identified in the *CAN1* ORF isolated from ethanol-treated cells. In line with our previous observations (Fig. 2 and Supplementary Fig. 2 and 3), this could again indicate that, in our conditions, acetaldehyde is not the main mediator of ethanol-induced mutagenesis.

A more in-depth look at the specific type of mutations revealed that cells exposed to 6% ethanol display different types of nucleotide substitutions, with more C to T substitutions compared with 0% ethanol (Fig. 3c, two-sided Fisher's test, $p = 0.0322$). This type of transition has been linked to spontaneous deamination of cytosine nucleotides. These transitions could indicate the presence of increased levels of single-stranded DNA in ethanol-exposed cells, as ssDNA is more prone to spontaneous deamination of cytosine residues[49].

Hence, we next checked for the formation of ssDNA upon exposure to ethanol. We quantified relocalization of the yellow

fluorescent protein (YFP)-tagged ssDNA-binding Rfa1 protein (Rfa1-YFP) to foci as a measure for ssDNA[50]. Cells not exposed to ethanol display mostly a diffuse nuclear localization of Rfa1, consistent with no ssDNA accumulation (Supplementary Fig. 7a). When cells go through S phase in the presence of methyl methanesulfonate (MMS) (a DNA alkylating agent known to cause the formation of ssDNA), cells progressively accumulate bright Rfa1 foci (Supplementary Fig. 7c). However, when cells go through S phase in the presence of 6% ethanol, we do not observe an increase of Rfa1 foci (Supplementary Fig. 7b), indicating that ethanol-exposed cells do not contain more ssDNA regions than untreated cells.

**Ethanol causes proteotoxic stress**. To learn more about the cellular response to ethanol, we performed RNA sequencing (RNA-seq) of ethanol-exposed cells. The results hinted at replication stress in ethanol-exposed cells. For example, *RNR3* expression levels are increased in ethanol-exposed cells (Supplementary Data 2). *RNR3* encodes a subunit of ribonucleotide reductase, an enzyme catalyzing the rate-limiting step in dNTP synthesis. *RNR* expression level is a key transcriptional read-out for replication stress and is highly induced under replication stress conditions[51].

A broader analysis of differentially regulated genes revealed that genes involved in protein degradation and refolding are significantly upregulated in ethanol-exposed cells (Fig. 4a and Supplementary Data 3). Ethanol has been reported to cause denaturation of proteins and the formation of protein inclusions[52]. In our RNA-sequencing data, we find evidence that exposure to 6% ethanol, a relatively low, naturally occurring, and sublethal concentration, also causes proteotoxic stress. For example, *KAR2* is upregulated 12-fold in cells that have been exposed to ethanol for two generations (Supplementary Data 2). *KAR2* encodes a molecular chaperone whose transcript level increases in response to high levels of unfolded proteins[53]. Genes encoding other chaperones, such as *HSP82* and *HSP104*, are also upregulated in ethanol, further suggesting that ethanol causes protein misfolding (Supplementary Data 2).

Subsequent experiments confirmed that ethanol causes proteotoxic stress. First, we used a well-established read-out for protein aggregation, namely aggregation of a fluorescently tagged reporter, the human Von Hippel-Lindau (VHL) protein[54]. When cells accumulate misfolded proteins, the protein quality control machinery becomes overloaded and the VHL protein forms aggregates, visible as fluorescent foci[54]. Ethanol causes an increase in such fluorescent foci, further corroborating the hypothesis that ethanol causes proteotoxic stress (Fig. 4b). In conditions where many unfolded proteins accumulate, proper proteasomal functioning is crucial for clearing these misfolded proteins, to safeguard the cell against proteotoxic stress. Proteasome inhibition by the drug MG-132 severely inhibited growth in 6% ethanol, indicating that proper proteasomal functioning is required in these conditions (Fig. 4c).

Together, these analyses show that a key marker for replication stress is upregulated, and that cells are experiencing proteotoxic stress, likely related to misfolding of proteins, when exposed to ethanol. This proteotoxic stress could even ultimately lead to replication stress, as it could result in unstable replisomes caused by, e.g., misfolding of crucial replisome components.

**Ethanol slows down replication**. To further check whether ethanol causes replication stress, we investigated Sml1 levels in ethanol-exposed cells (Fig. 5a). Sml1 is an established sensor for Mec1 checkpoint activation. Mec1 is a genome integrity checkpoint protein kinase that becomes activated when cells experience

replication blockages[55] or DNA damage. Sml1 is a downstream target of Mec1 whose levels decrease when the replisome encounters replication blockages or DNA damage[56]. Ethanol-exposed cells display decreased Sml1 levels, indicative of replication stress checkpoint activation (Fig. 5a).

Interestingly, although quantification of YFP-Rnr3 fluorescence signal in ethanol-exposed cells are in line with our RNA-seq data, namely that ethanol significantly increases Rnr3 protein levels, exposing cells to MMS causes a much higher increase in Rnr3 protein levels (Fig. 5b). In addition, Rad53, the effector kinase that is phosphorylated after activation of either the DNA damage checkpoint or the DNA replication checkpoint, is not phosphorylated in response to EtOH treatment (Supplementary Fig. 8). We also do not detect a significant increase in ssDNA levels in ethanol-exposed cells (Supplementary Fig. 7), with ssDNA accumulation being a signal for checkpoint activation. Together, this data indicate that ethanol only causes a mild replication stress checkpoint activation, one that is much less pronounced compared with MMS.

We next assessed the effects of ethanol exposure on replication by examining cell cycle progression (Fig. 5c and Supplementary Fig. 9). Cells were synchronized in $G_1$ phase with α-factor and released into S phase in the presence of 0 or 6% ethanol. S-phase progression was delayed in cells exposed to 6% ethanol, suggesting a replication defect in ethanol-exposed cells.

To further confirm whether DNA replication is affected upon ethanol exposure, we analyzed replication progression by DNA combing from asynchronous cell cultures. This allowed us to measure replication fork progression (track length). Ethanol-exposed cells showed significantly shorter track length compared with cells that were not exposed to ethanol (Fig. 5d, e). This data indicates that the cell cycle delay observed in ethanol-exposed cells is caused by altered DNA replication and suggest that ethanol affects replication fork progression.

**EtOH affects Mrc1 localization**. As replication rate is highly dependent on Mrc1, the homolog of metazoan Claspin[30,57,58], an evolutionary conserved component of the replisome that links the replicative helicase with DNA polymerase activities[59], we next investigated the effects of ethanol on Mrc1. Interestingly, ethanol causes relocalization of Mrc1—from the replication fork to a perinuclear compartment, termed IntraNuclear Quality Control Compartment (INQ) (Fig. 6a, b). INQ has been shown to sequester misfolded, ubiquitylated, and sumoylated proteins in response to proteotoxic, replication, and genotoxic stresses[60].

As Mrc1 is important to maintain a normal replication rate, efficient DNA replication and replisome stability[30,58,59], and as defective replisomes have been linked to increased genome instability[27,61], our data suggest that ethanol could affect mutation rate by dissociating Mrc1 from the replisome. Mrc1 relocalization in ethanol conditions could also explain the DNA replication defect observed in ethanol-exposed cells, as exposing cells to ethanol for 2 h affects both Mrc1 localization and replication rate (Fig. 5).

To test whether the ethanol-associated increase in mutation rate is affected by Mrc1, we determined mutation rate of a strain overexpressing *MRC1* in 0 and 6% ethanol conditions (Fig. 6c). *MRC1* overexpression greatly reduces the mutagenic effect of ethanol. In summary, our data shows that Mrc1 relocalizes to INQ in response to ethanol, with this relocalization potentially affecting replisome stability, replication rate and genome stability.

**Mutagenic effect of ethanol requires error-prone polymerases**. Replication forks lacking Mrc1 progress more slowly and have

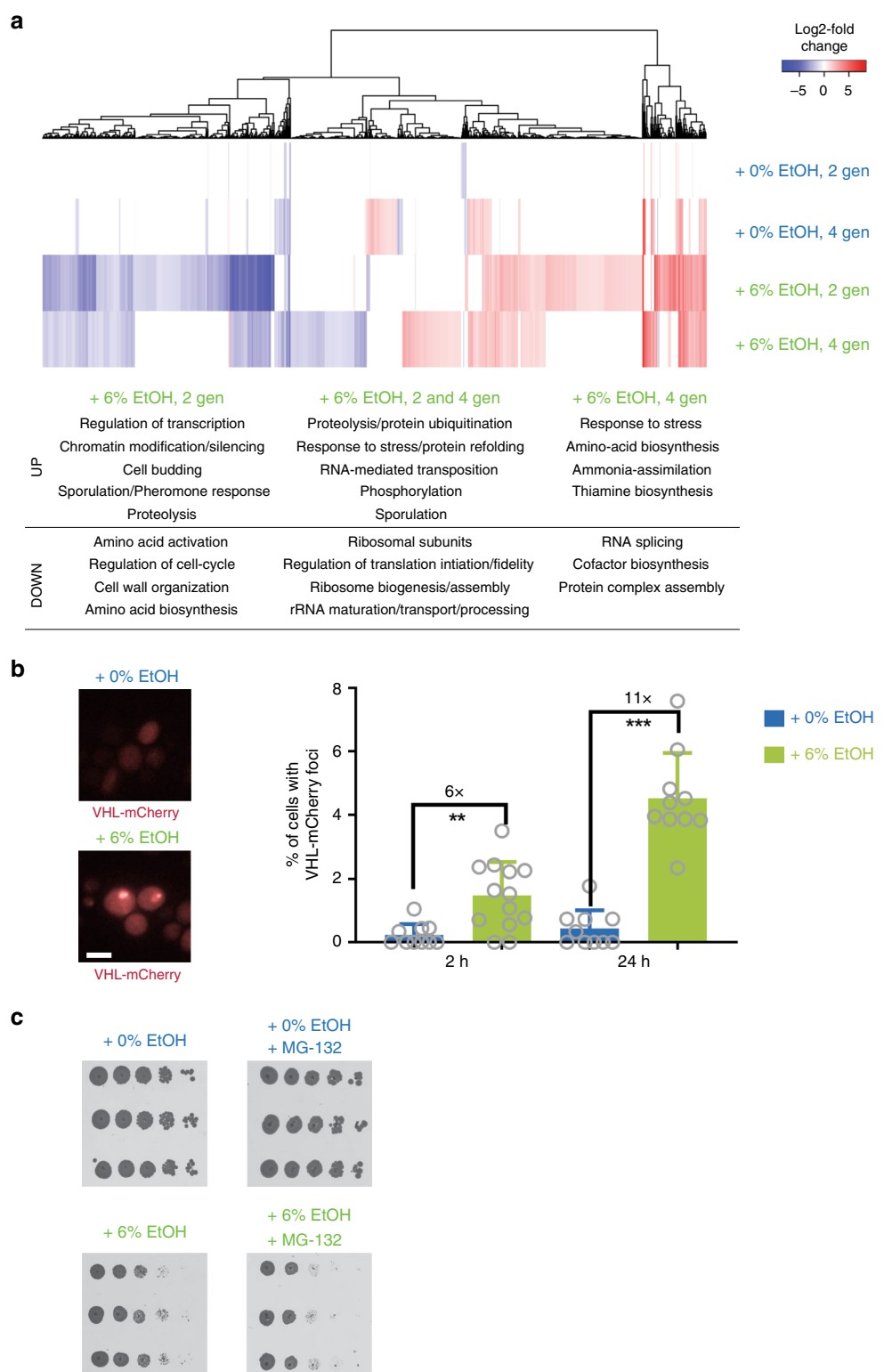

been reported to lead to DNA damage[30,57,58,62]. Translesion polymerases are recruited to sites of replication fork stalling and/ or DNA damage[27]. Translesion polymerases have a higher error-rate than the regular replicative polymerases and are hence sometimes referred to as error-prone polymerases[16]. These polymerases also often make consecutive errors in a single round of DNA synthesis[63], causing a typical pattern of complex muta-tions—defined as multiple mutations that are separated from their nearest neighbor by no more than a stretch of ten

nucleotides[29]. Although not statistically significant, we also observe such complex mutations in our ethanol-exposed cells (Fig. 7a).

We used multiple experimental approaches to investigate whether error-prone polymerases are involved in the ethanol-associated increase in mutation rate. First, stalling of replication forks triggers ubiquitination of PCNA/Pol30, the highly con-served sliding clamp for replicative polymerases[64]. Ubiquitin attachment in turn acts as a scaffold for other proteins involved in

**Fig. 4 Ethanol causes proteotoxic stress. a** RNA-seq indicates that ethanol causes proteotoxic and replication stress. Cells (VK111) were grown for two and four generations (indicated as gen) in medium supplemented with the indicated ethanol concentrations. Heat map represents expression changes (expressed as log2 fold change) in the different samples relative to time point zero. Values are colored according to the scale shown, with blue indicating low values and red indicating high values. Samples are hierarchically clustered based on expression changes. Term-enrichment analysis of differentially expressed genes indicates that processes involved in protein degradation and refolding are significantly upregulated in ethanol conditions (see also Supplementary Data 3 for enrichment scores and *p*-values). **b** Ethanol exposure causes aggregation of VHL-mCherry. Cells expressing VHL-mCherry (VK3703) were grown to mid-exponential phase in synthetic medium lacking uracil with 2% raffinose as a carbon source. VHL-mCherry expression was induced by addition of 3% galactose (final concentration) for 3 h, after which cells were exposed to the indicated ethanol concentrations for the indicated times. At least 1500 cells (from 2 independent biological replicates) were analyzed per condition. For 2 h in 0%, 2397 cells were analyzed. For 2 h in 6%, 2363 cells were analyzed. For 24 h in 0%, 1770 cells were analyzed. For 24 h in 6%, 1917 cells were analyzed. Scale bar represents 5 μm. Bars represent average ± SD. Statistical significance was assessed using a two-tailed unpaired *t*-test with Welch's correction. **P < 0.01, ***P < 0.001. Specifically, for 2 h, $p = 0.0013$ and for 24 h, $p < 0.0001$. **c** Inhibition of proteasome activity reduces growth in 6% ethanol. Tenfold serial dilutions of WT strain VK111 (start $OD_{600nm} = 0.1$) were spotted on YPD plates containing 0 or 6% ethanol with or without 100 μM MG-132. YPD plates are shown after 2 days of incubation; YPD ethanol plates are shown after 3 days of incubation. Source data for this figure are provided as a Source Data file.

replication past difficult or damaged DNA regions. Specifically, mono-ubiquitination of Pol30 at residue K164 acts as a docking site for error-prone polymerases[64,65]. To test whether Pol30 ubiquitination is required for the mutagenic effect of ethanol, we determined the mutation rate of a strain containing a Pol30[K164R] variant that cannot be ubiquitinated (Fig. 7b). Interestingly, mutating this ubiquitination site completely abolishes the mutagenic effect of ethanol.

Next, we created single knockouts of the different error-prone polymerases and determined mutation rate in 0 and 6% ethanol (Fig. 7b). *S. cerevisiae* contains four translesion polymerases: polymerase ζ consists of Rev3 and Rev7, with *REV3* encoding the catalytic subunit and *REV7* the accessory subunit that stimulates Rev3 activity; polymerase η consists of Rad30, whereas Rev1 is a deoxycytidyl transferase that can form a complex with Rev3 and Rev7. Deletion of *REV1* or *REV7* almost completely abolishes the effect of ethanol on mutation rate, whereas *rev3Δ* and *rad30Δ* strains show a modest reduction in ethanol-associated mutation rate increase, indicating that polymerase ζ is the primary source of ethanol-induced mutations.

Lastly, we examined the presence of Rev1-3-7 and Pol2, the catalytic subunit of replicative polymerase ε, at replication forks using Chromatin immunoprecipitation (ChIP) (Fig. 8). Cells containing epitope-tagged versions of these proteins were arrested with mating factor and released into medium containing either 0 or 6% ethanol. At ARS305, an early-firing origin, Pol2 is loaded 20 min after release and arrives 3 kb upstream and downstream 20–60 min later. In the presence of ethanol, less Pol2 is bound to chromatin, suggesting that the replisome is unstable[66]. We also observe a general tendency for more translesion synthesis polymerase binding to chromatin in ethanol. This further supports our model that ethanol causes replication stress and triggers recruitment of error-prone polymerases.

Taken together, our results show that the mutagenic effect of ethanol is caused by the recruitment of translesion polymerases to replicating DNA. The data support a model where ethanol exposure causes general protein instability and triggers relocalization of Mrc1 from the replication fork to INQ. In the absence of Mrc1, the replisome becomes unstable and progresses more slowly. This triggers the recruitment of error-prone polymerases, through the ubiquitination of Pol30, ultimately resulting in an increased mutation rate in the presence of ethanol (Fig. 9).

## Discussion
At high levels, ethanol is lethal to all living organisms. Our results show that even relatively low, naturally occurring ethanol levels lead to an increased mutation rate in the model eukaryote *S. cerevisiae*. Specifically, we show that ethanol causes both proteotoxic and replication stress, and that ethanol exposure results

in relocalization of Mrc1, a crucial replisome component, from the replication fork to INQ. This triggers exchange of the replicative polymerase with error-prone polymerases, which ultimately leads to increased mutation rates.

Currently, the primary trigger for Mrc1 relocalization is still unkown. Genotoxic, proteotoxic and replication stress can cause Mrc1 relocalization to INQ[60]. Further research is required to determine whether Mrc1 is, for example, recruited to INQ as part of a general, proteotoxic response (with Mrc1 being just one of the proteins that is misfolded and targeted to INQ), or if Mrc1 relocalization is part of a specific signaling pathway[67].

Furthermore, our experiments are not conclusive about the potential role of acetaldehyde. However, it is clear that the mechanisms underlying the mutagenic effect of ethanol are more complex than previously thought. Ethanol and/or acetaldehyde could cause chemical damage to the DNA, which causes replication fork stalling and recruitment of error-prone polymerases. It seems equally plausible that the recruitment of these polymerases is directly caused by the proteotoxic effect of ethanol on the replication fork, causing it to become unstable and stall. In fact, both mechanisms are not mutually exclusive and are difficult to disentangle. In the case of lesions due to chemical DNA damage (e.g., caused by acetaldehyde-derived adducts), we would expect replication fork collapse and a strong checkpoint activation. Interestingly, we do not observe a strong checkpoint activation by ethanol, indicating that replication forks do not collapse, again pointing to the complex mechanisms underlying the mutagenic effects of ethanol.

The proteotoxic stress observed in ethanol-exposed cells could be due to ethanol-generated protein adducts. In fact, studies have identified various ethanol-induced protein adducts[7,68]. These adducts have been mainly attributed to ethanol metabolism, with acetaldehyde and ROS reacting with proteins to form adducts. Although our data seems to indicate that ROS and likely also acetaldehyde are not responsible for the mutagenic effect of ethanol, it is possible that ethanol-generated adducts, perhaps together with other sources of ethanol-derived proteotoxic stress, such as denatured proteins, could underlie the observed mutagenic effect of ethanol, potentially by affecting replication fork components.

Different environmental stresses can trigger a common transcriptional response, the environmental stress response, comprising changed expression of >300 genes[69]. Such sudden, massive transcriptional changes can lead to genome instability, caused by replication–transcription conflicts[70]. Mrc1 plays a key role in preventing these replication–transcription conflicts upon sudden stress: stressors such as heat, oxidative and osmotic stress, and low glucose levels trigger N-terminal phosphorylation of Mrc1 and a subsequent delay in replication[67,71]. This delay

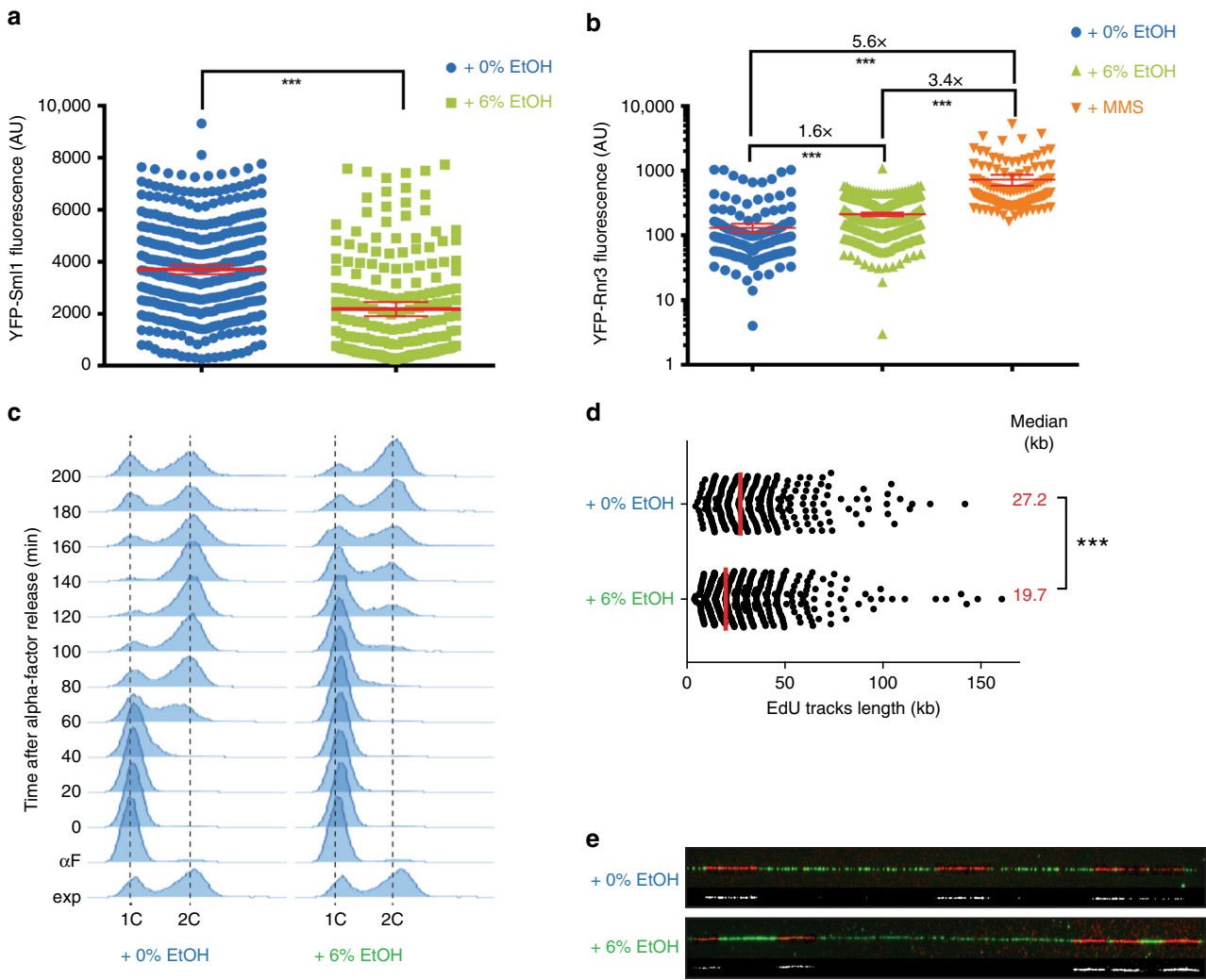

**Fig. 5 Ethanol affects replication rate. a** Ethanol weakly activates the replication checkpoint. Cells expressing YFP-Sml1 (IG101-12D) were exposed to 0 or 6% EtOH for 2 h and imaged using fluorescence microscopy. Ethanol-exposed cells display decreased Sml1 levels. Data represent average fluorescence intensities of individual cells. Error bars represent 95% confidence intervals. $N = 471$ cells for 0% ethanol and $N = 198$ cells for 6% ethanol. Statistical significance was assessed using a two-tailed unpaired $t$-test with Welch's correction. ***$P < 0.001$. Specifically, $p$-value $< 0.0001$. AU, arbitrary units. **b** Ethanol increases cellular Rnr3 protein levels. Cells expressing YFP-Rnr3 (W6986-1B) were exposed to 0 or 6% EtOH for 2 h or 0.03% MMS for 1 h and imaged using fluorescence microscopy. $N = 241$, 374, and 123 for 0% ethanol, 6% ethanol, and MMS, respectively. Data represent average fluorescence intensity; error bars represent 95% confidence intervals. Statistical significance was assessed using a two-tailed unpaired $t$-test with Welch's correction. ***$P < 0.001$. Specifically, $p$-value $< 0.0001$. AU, arbitrary units. **c** Cell cycle progression is slower in ethanol-exposed cells. Wild-type cells were arrested in $G_1$ with α-factor and were released synchronously into S phase with the addition of pronase, in medium containing 0 or 6% ethanol. Cells were collected at the indicated time points. DNA content was assessed using flow cytometry after PI staining. The 1C peak corresponds to cells in the $G_0/G_1$ phase. The 2C peak corresponds to cells in the $G_2/M$ phase. **d, e** Replication fork progression is perturbed by ethanol. Fork speed, measured after pulse incorporation of EdU and DNA combing, was analyzed in asynchronous cell cultures (strain PP2226) exposed for 2 h to 0 or 6% ethanol. For each condition, at least 329 tracts coming from 3 independent replicates were analyzed. $N = 329$ and 637 for 0 and 6% ethanol, respectively. The scatter dot plot depicts the distribution of EdU track lengths. Medians are shown by a red line and are indicated as data labels (red). Statistical significance was assessed using a two-tailed Mann–Whitney unpaired non-parametric $t$-test. ***$P < 0.001$. Specifically, $p$-value $< 0.0001$. **e** Examples of the DNA fibers (green) containing EdU tracks (red). EdU tracks are highlighted in white below each fiber. Source data for this figure are provided as a Source Data file.

prevents replication–transcription conflicts and genome instability. Although we currently do not know whether ethanol also triggers Mrc1 phosphorylation, our data provide another argument for Mrc1 being a key factor in controlling genome stability in different stresses.

Replication forks lacking Mrc1 progress more slowly and have been reported to lead to DNA damage[30,57,58,62]. PCNA/Pol30 becomes ubiquitinated in the presence of stalled replication forks and DNA damage, and this ubiquitination triggers recruitment of error-prone polymerases[64,65]. Error-prone polymerases are indeed required for the ethanol-associated increase in mutation

rate and mutating the ubiquitination site in PCNA/Pol30 abolishes the mutation increase observed under ethanol conditions. Hence, our study provides a plausible mechanistic explanation for the observed mutation rate increase in ethanol-exposed cells. Consistent with a previous report that DNA synthesis by error-prone polymerases does not require checkpoint signaling[72], we do not observe checkpoint activation by ethanol.

Increasing mutation rates can lead to increased genetic variability, which has been proposed to be beneficial in some cases, as the increased mutation rate might increase the chances of evolving better adapted mutants. However, the selective pressures

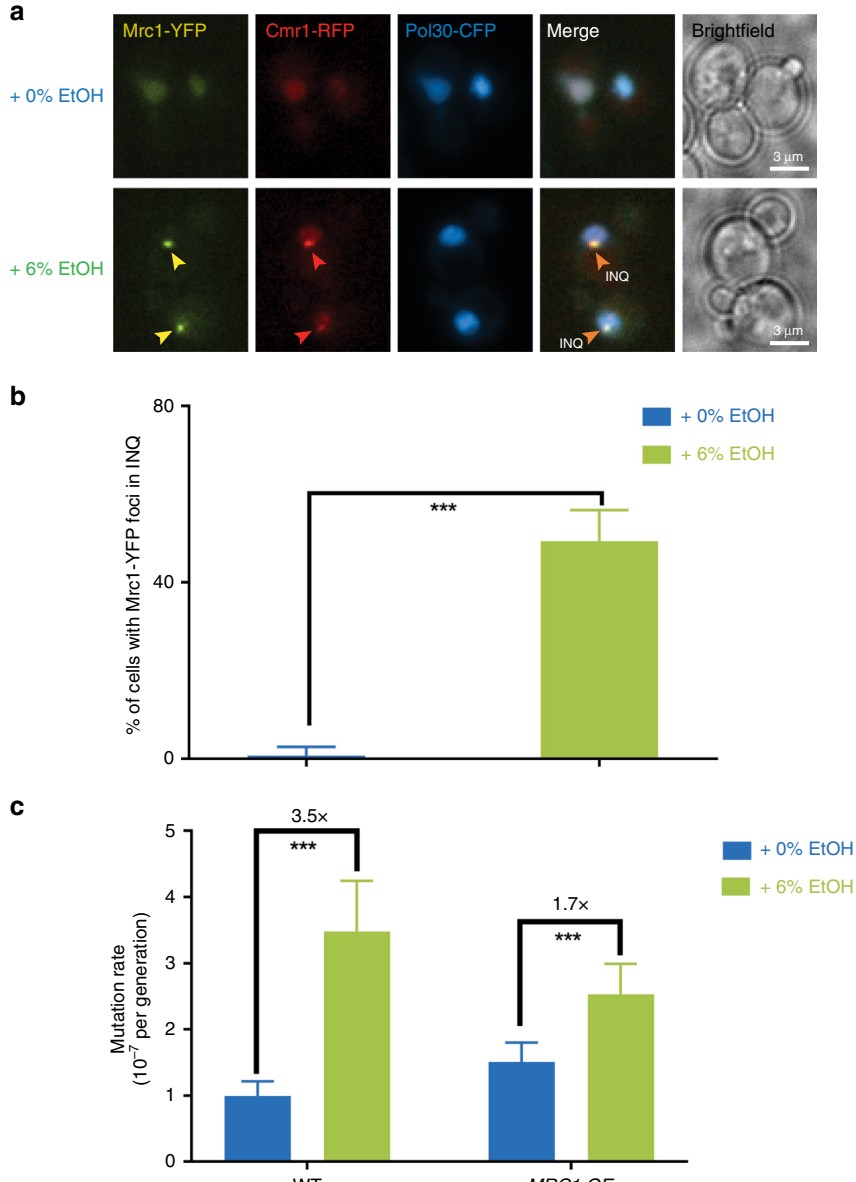

**Fig. 6 Ethanol affects the replisome. a** Mrc1 relocalizes to INQ in ethanol-exposed cells. Ethanol redistributes Mrc1 (yellow) from the replication fork to INQ. Cells expressing Mrc1-YFP, Cmr1-RFP, and CFP-Pol30 (CC71-34B) were exposed for 2 h to either 0% or 6% ethanol and imaged using fluorescence microscopy. Representative images are shown. Cmr1 (red) is an established marker for INQ, whereas Pol30 (blue) is a marker for the replication fork. Scale bar represents 3 μm. **b** Mrc1 mostly localizes to INQ in 6% ethanol conditions. Quantification of Mrc1 foci in strain CC71-34B. $N = 431$ and 228 for 0% and 6% ethanol, respectively. Bars represent percentage of cells containing foci; error bars represent 95% confidence intervals. Statistical significance was assessed using a one-tailed Fisher's exact test. ***$P < 0.001$. Specifically, $p = 0.00025$. **c** *MRC1* overexpression reduces ethanol-associated increase in mutation rate. Cells (VK111 and VK3761) were grown in synthetic media (2% glucose) at different ethanol concentrations (v/v). For each condition, 54 cultures were analyzed. Data represent mutation rate estimates, as determined by fluctuation assays on canavanine. Error bars represent 95% confidence intervals. Statistical significance of differences in mutation rates was assessed using a likelihood ratio test. ***$P < 0.001$. Specifically, $p$-values are $1.5668 \times 10^{-7}$ and $7.24 \times 10^{-5}$ for WT and *MRC1* OE, respectively. Source data for this figure are provided as a Source Data file.

causing SIM and the implications of SIM for evolutionary dynamics are hotly debated[73–75]. Some researchers consider SIM an adaptive strategy, that effectively increases the supply of mutations needed to overcome adaptive hurdles; whereas others consider it as an unavoidable byproduct of specific mechanisms and proteins, including error-prone polymerases, which are induced or recruited under stress. Most non-neutral mutations are deleterious and so it is still debated whether there could be an evolutionary advantage to temporarily increasing mutation rates.

*S. cerevisiae* has proven a useful model system for higher eukaryotes. Although caution is needed, it is tempting to correlate our observations in yeast with observations in humans. Ethanol is listed as a class I carcinogen by the World Health Organization and yearly around 800,000 cases of cancer are linked to patient alcohol consumption[76]. Although epidemiological data clearly demonstrated a link between ethanol intake and tumor formation, the precise mutagenic mechanism of ethanol in humans is poorly studied. Interestingly, several mutational signatures have recently been identified in esophageal squamous cell carcinoma, a cancer with a clear link to ethanol exposure. Multiple signatures correlate with the patient's alcohol consumption, although only one could be explained by acetaldehyde[77]. This again highlights

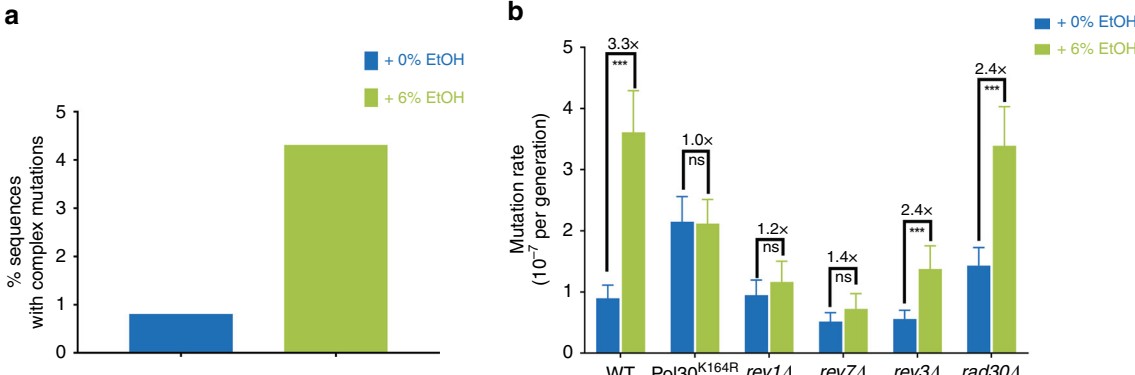

**Fig. 7 Mutagenic effect of ethanol depends on error-prone polymerases. a** A trend for more complex mutations in ethanol-exposed cells. Complex mutations are defined as multiple mutations within ten nucleotides. For 0% EtOH, a total of 121 colonies were analyzed; for 6% EtOH, a total of 113 colonies were analyzed. **b** Ethanol-associated mutagenesis depends on error-prone polymerases and ubiquitination of Pol30 at Lys164. Cells (VK111, VK3831, VK3548, VK3614, MVP1101, and MVP1105) were grown in synthetic media (2% glucose), supplemented with the indicated ethanol concentrations (v/v). For each strain and condition, 54 cultures were analyzed. Data represent mutation rate estimates, as determined by fluctuation assays on canavanine. Error bars represent 95% confidence intervals. Statistical significance of differences in mutation rates was assessed using a likelihood ratio test. ***$P < 0.001$, ns nonsignificant. Specifically, p-values are 0 for WT, $9.99 \times 10^{-1}$ for pol30 mutant, 0.25168 for $rev1\Delta$, 0.1036 for $rev7\Delta$, $1.8445 \times 10^{-6}$ for $rev3\Delta$ and $1.1643 \times 10^{-9}$ for $rad30\Delta$. Source data for this figure are provided as a Source Data file.

that other mechanisms, apart from ethanol conversion to acetaldehyde, may underlie alcohol-related carcinogenesis. Although the exact involvement of acetaldehyde in ethanol-associated genome instability warrants further investigation, our findings suggest a new model for ethanol-related genome instability (Fig. 9). Ethanol affects highly conserved proteins and processes as follows: (i) Mrc1 is an evolutionary highly conserved replisome component, called Claspin in higher eukaryotes. Claspin is also required for normal DNA replication and replisome stability[78]; (ii) proteotoxic stress induces INQ-like structures in HeLa cells[60]; (iii) dysfunctional replication forks and replication stress have been linked to increased genome instability in tumors[26]; and (iv) error-prone polymerase-associated mutations have been found in alcohol-associated tumors[17]. Taken together, these similarities make it tempting to speculate that ethanol could affect similar processes in mammalian cells.

Our data links error-prone polymerases to alcohol-related mutations in *S. cerevisiae*. Interestingly, a recent study reported error-prone polymerase-associated mutational spectra in alcohol-related tumors[17]. In the latter case, tumor samples displayed a mutational spectrum characteristic for PolH (encoded by *RAD30* in *S. cerevisiae*), whereas our data implicate PolZ as the primary source of alcohol-related mutations in *S. cerevisiae*.

Understanding the precise mutagenic effects of ethanol in yeast may open up new routes to evaluating and reducing the health risks of ethanol in humans.

## Methods

**Strains used in this study**. Strains used in this work were derived from a prototrophic haploid S288c, unless otherwise indicated. A full list of strains used, with their complete genotype, can be found in Supplementary Table 2. Primers used for strain construction and verification were ordered from IDT or TAG Copenhagen; primer sequences are listed in Supplementary Table 3.

Deletion strains were generated by amplifying the *HygB* cassette (pCB1) or the *KaNMX* cassette (pUG6) from plasmids using primers (Supplementary Table 3), which contained at least 40 bp sequence homologous to target DNA. The PCR product was then used for directed integration of the cassette and replacement of the target locus. Yeast transformation was carried out using the LiAc procedure. Transformants were verified by PCR using specific primers (Supplementary Table 3). To obtain mutants showing increased *ADH2* or *MRC1* expression, we integrated a modified *TEF1* promoter directly upstream of the respective ORF. To introduce mutations in *POL30* and *SSD1*, we used a CRISPR-Cas9 based protocol, see below. Strains expressing FLAG-tagged Rev1, Rev3, and Rev7 from their endogenous loci were constructed by amplifying a 3xFLAG::HIS3 cassette from plasmid pBP81 using primers with 40 bp 5′ homology arms to facilitate integration

immediately before the STOP codon of the gene. Constructs were verified by sequencing.

**Plasmids**. A full list of plasmids used can be found in Supplementary Table 4.

**Media**. Media used in this study consisted of 1% yeast extract, 2% peptone, and 2% glucose (YPD). Plates of these media were made with 1.5% agar for standard growth conditions. YPD containing Hygromycin B (Invitrogen) (200 mg/L), G418 (Formedium) (200 mg/L), or ClonNat (100 mg/L) were used for selection of yeast transformants. Synthetic complete media consisted of 6.7 g/L Yeast Nitrogen Base with ammonium sulfate and without amino acids, 1.77 g/L CSM-Ura (Formedium), 50 mg/L uracil (Sigma), and 2% glucose. Canavanine plates consisted of 6.7 g/L Yeast Nitrogen Base with ammonium sulfate and without amino acids, 0.74 g/L CSM-Arg (Formedium), 60 mg/liter L-canavanine (Sigma-Aldrich), 1.5% agar, and 2% glucose. FOA plates consisted of 0.67% Yeast Nitrogen Base with ammonium sulfate and without amino acids, 0.2% CSM-Ura (Formedium), 50 mg/L uracil (Sigma-Aldrich), 1 g/L FOA (Formedium), 1.5% agar, and 2% glucose. Ethanol and methanol were purchased from VWR in their highest purity; isopropanol was purchased from Sigma-Aldrich.

**CRISPR-Cas mutation of *POL30* and *SSD1***. The 5′-phosphorylated oligonucleotide sequences that served as guide sequence were ordered from IDT and are listed in Supplementary Table 3. To create annealed guide sequence-containing primers, 100 μM of each primer was added to a PCR tube, were incubated in a thermocycler for 5 min at 95 °C, and subsequently cooled to 16 °C, at the slowest ramp rate of the thermocycler. Next, annealed guide sequence-containing primers were ligated into calf intestinal phosphatase-treated, BsmBI-digested parent vector pV1382. This was transformed into chemically competent *E. coli* cells. Plasmids were extracted and verified by Sanger sequencing and restriction digest. Repair templates were generated with 60-bp oligonucleotide primers (IDT) containing 20 bp overlaps at their 3′-ends centered at the mutation point. Primers were extended by thermocycling performed with *Ex Taq* (TAKARA) and double-stranded products were purified directly from this PCR reaction using Qiaquick Gel Extraction Kit (Cat# 28706) following the manufacturer's instructions. Transformations were performed with 0.3–5 μg of plasmid DNA and with 1–5 μg of repair template (where applicable). Transformation was done using the standard lithium acetate protocol, and suspensions were plated on selective media. Presence of the mutation was verified by Sanger sequencing.

**Fluctuation assay**. Fluctuation assays to determine mutation rates were performed as follows: precultures were grown overnight at 30 °C in Synthetic Complete (SC)-Arg. Cells were subsequently brought to a starting density of 5000 cells/mL in SC glucose medium containing either 0 or 6% ethanol. Cultures (200 μl) of these cells were grown at 30 °C in synthetic complete medium containing 2% glucose and the indicated ethanol concentration in 96-well plates for 48 h. To prevent ethanol evaporation, outer wells only contained medium and 96-well plates were sealed with an adhesive seal and plastic lid, and plates were wrapped in parafilm. For each genotype, at least 60 replicates were grown. Fifty-four cultures were then plated on canavanine plates (containing 60 mg/L L-canavanine sulfate (Sigma-Aldrich)) for mutant selection and incubated at 30 °C for 3 days, after which colonies were

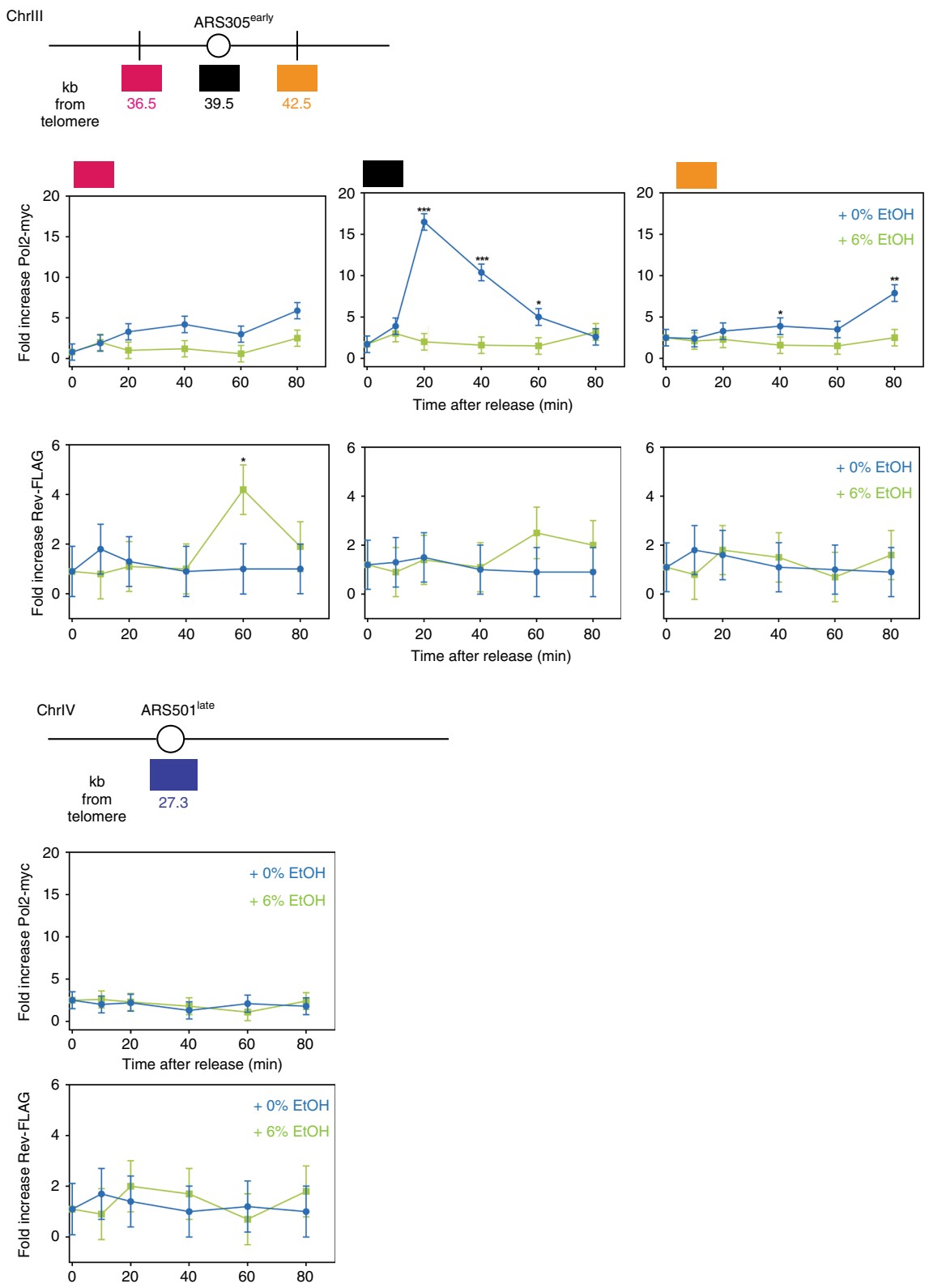

counted. Six cultures were diluted and plated on YPD to determine total cell counts. Fluctuation assays using *URA3* as a mutation reporter were performed similarly, except that cultures were plated on FOA (Sigma-Aldrich) containing plates for mutant selection and incubated at 30 °C for 2 days, after which colonies were counted.

Colony counts on selective and non-selective plates were used to estimate mutation rates using the rSalvador R package (v1.7)[79]. Likelihood ratio methods, as implemented in rSalvador, were used to calculate 95% confidence intervals, as well

as to assess the statistical significance of differences between mutation rate estimates.

**can^R mutant frequency after short ethanol exposure**. Precultures were grown as for standard fluctuation assay. Cell count of precultures was determined using BioRad CellCounter and 10^5 cells per culture were used to set up the experiment. Cells were incubated for 2 h in the presence of 0 or 6% ethanol and then plated on

**Fig. 8 Replication in ethanol-exposed cells switches to error-prone polymerases.** Schematic representation of genomic regions containing the early-firing origin ARS305 and late-firing origin ARS501. Boxes indicate qPCR amplicons location at ARS305 (black box), an early-firing origin, as well as non-origin sites −3 kb (pink box) and +3 kb (yellow box) from ARS305. Primers for the late-firing origin ARS501 are used to monitor late origin activation and background signals. Origins are described in refs. [83,84]. ChIP was performed as described in "Methods" on myc-tagged Pol2 and FLAG-tagged Rev1-3-7. G$_1$-arrested cells (strain ML996-2D) were released into media containing 0 or 6% ethanol for the times indicated. Values are average of three biological replicates, with two technical repeats each; error bars represent SD. Data represent the real-time PCR signal as fold increase of the IP over the beads-only control for Pol2 and Rev1-3-7. Statistical significance was assessed using a two-tailed unpaired $t$-test with Welch's correction. *$P < 0.05$, **$P < 0.01$, ***$P < 0.001$. Specific $p$-values can be found in Supplementary Table 1. Source data for this figure are provided as a Source Data file.

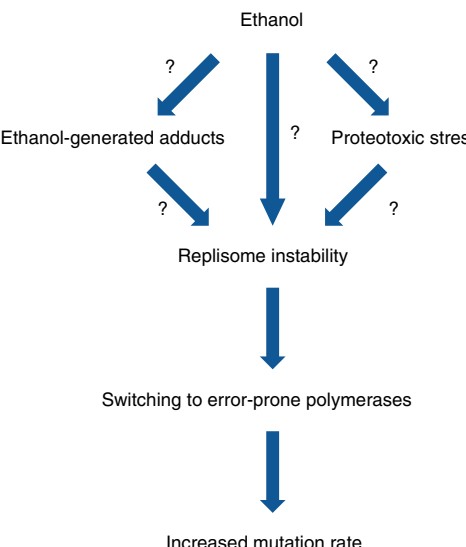

**Fig. 9 Model for effect of ethanol on genome stability.** Proposed model for the mutagenic effect of ethanol. See text for more details.

canavanine plates. Resistant colonies were counted after 3 days incubation at 30 °C and $can^R$ mutant frequency was obtained by dividing the number of resistant colonies by the total number of cells per culture plated.

**$can^R$ mutant frequency after acetaldehyde exposure**. This protocol is modified from ref. [80]. Briefly, cells were grown as for standard fluctuation assay. Pre-chilled (−20 °C) acetaldehyde (Sigma) was added via an ice-cold plastic syringe into ice-cold rubber-sealed glass vials containing cultures (final cell density 5000 cells/mL). Cultures were incubated for 30 min at 4 °C, followed by a 2 h incubation at 30 °C. Cultures were spun down, resuspended in regular SC medium, and incubated for 24 h at 30 °C before plating on canavanine plates. Resistant colonies were counted after 3 days incubation at 30 °C and $can^R$ mutant frequency was obtained by dividing the number of resistant colonies by the total cell count.

**Identifying mutations at *CAN1* locus**. The *CAN1* locus was amplified using ExTaq from genomic DNA isolated from canavanine-resistant colonies using a standard zymolyase protocol. The resulting PCR product was subsequently sent for Sanger sequencing by VIB Genetic Service Facility. The sequence and quality of the reads were extracted to fastq format from the sequencing chromatogram files using EMBOSS seqret version 6.6.0 and the ends were quality trimmed with a quality cutoff of Q20 using fastq_quality_trimmer from the FASTX-Toolkit (version 0.0.14). The sequences from each segment were aligned using MUSCLE version 3.8.31. The aligned segments were then each aligned to the *CAN1* gene from S288c and combined in AliView version 1.18.1. The alignment was then checked and manually curated in AliView. The reads for each individual colony were extracted from the alignment and were joined into single sequences at overlaps. In cases of sequence disagreement, preference was given to a base if it agreed with S288c *CAN1*, otherwise IUPAC ambiguity codes were introduced. Ns were added in the case of missing data. Each colony sequence was then compiled into a final multiple sequence alignment. Mutations were analyzed by a perl script, which categorized them into indels or silent/missense/nonsense mutations, and checked whether they were transitions/transversions. All canavanine-resistant colonies carried at least one mutation in the *CAN1* gene. Sequencing data have been deposited at GenBank, accession codes MT509124–MT509357.

**Determination of viability**. Cellular viability was determined using methylene blue staining protocol. Briefly, cells were grown exactly as for performing fluctuation

assays. Cultures were diluted 1/100 in SC medium and 100 µL of a filter sterilized 0.1% (w/V) methylene blue solution (Sigma-Aldrich) was added to 100 µL of this diluted culture. Samples were incubated at room temperature for 1 min and subsequently loaded on a counting slide (KOVA Glasstic Slide 10). Cells were visualized by microscopy and counted. Blue cells were considered dead and unstained cells viable. At least three independent cultures were counted for each strain and condition, with a minimum of 600 cells counted in total for each strain.

**Enzymatic measurements of acetaldehyde levels**. Acetaldehyde levels were determined enzymatically using a Megazyme Acetaldehyde Assay kit and following the manufacturer's instructions (Megazyme).

**HPLC measurements of acetaldehyde levels**. The acetaldehyde determination was performed as described by ref. [81]. Briefly, cells were grown to exponential phase (OD$_{600nm}$ = 0.2–0.3) in YPD/YPD 6% Ethanol. Approximately 1.2 mL of cell culture was withdrawn into 6 mL of pre-cooled (−40 °C) quenching and derivatization solution containing 0.9 g L$^{-1}$ 2,4-dinitrophenylhydrazine and 1% (v/v) phosphoric acid in acetonitrile. After mixing and incubating for 2 h on a shaking platform at 4 °C, samples were stored at −80 °C until further analysis. Prior to analysis, 1 mL of defrosted and well-mixed sample was centrifuged (15,000 × $g$, 3 min). The supernatant was analyzed via HPLC using a WATERS WAT086344 silica-based, reverse-phase C18 column operated at room temperature with a gradient of acetonitrile as a mobile phase. A linear gradient was generated from eluent A (30% (v/v) aqueous acetonitrile solution) and eluent B (80% (v/v) aqueous acetonitrile solution). The mobile-phase composition was changing from 0% to 100% of eluent B in 20 min, at a flow rate of 1 mL min$^{-1}$. A calibration curve was prepared with standard solution of 50.9 g L$^{-1}$ acetaldehyde-2,4-dinitrophenylhydrazine in acetonitrile.

**In vivo ROS measurements using H$_2$DCFDA**. Cells were grown as for standard fluctuation assay. H$_2$DCFDA (Sigma, stock concentration: 5 mg/mL in dimethylsulfoxide) was added to each culture to a final concentration of 12.5 µg/mL and light-protected samples were incubated for 30 min at 30 °C. Samples were divided over three Eppendorf tubes and incubated with nothing, 6% ethanol (final concentration), or 100 mM H$_2$O$_2$ for 2 h at 30 °C. Fluorescence was subsequently analyzed using Attune$^{TM}$ NxT Acoustic Focusing cytometer. A total of 50,000 cells was analyzed per sample and three independent cultures were analyzed for each treatment.

**RNA sequencing and analyses**. Illumina NextSeq 5000 paired-end reads (2 × 75 bp) were obtained by sequencing samples from the starting culture, and after two and four generations of growth in filter-sterilized YPD (4% glucose) medium containing either 0% or 6% (V/V) ethanol. All samples were taken in biological duplicates. Details on analysis can be found in Supplementary Information. The RNA-seq dataset generated and analyzed has been deposited in Sequence Read Archive, as Bioproject PRJNA632734 (http://www.ncbi.nlm.nih.gov/bioproject/632734).

**Spot assays**. Cultures were grown overnight to saturation in YPD. Tenfold serial dilutions (OD$_{600}$ of start dilution = 1.0) were spotted on agar plates. To test the effect of proteasome inhibition, cultures were also spotted on agar plates containing the proteasome inhibitor MG-132 (100 µM final concentration, Sigma-Aldrich)). Plates were sealed with parafilm and incubated at 30 °C. YPD plates were scanned after 2 days incubation and YPD Ethanol plates were scanned after 3 days incubation.

**Yeast live-cell imaging and fluorescence**. For detecting VHL fluorescent foci, cells containing plasmid pESC-mCherry-VHL encoding mCherry-tagged VHL were grown on SC-uracil containing 2% raffinose as a carbon source at 30 °C[54]. Saturated cultures were diluted into SC-uracil containing 3% galactose to induce mCherry-VHL expression. After 3 h induction, cultures were split in two and one part of the culture was incubated with 6% ethanol (final, V/V). mCherry-VHL fluorescence was imaged after 2 and 24 h incubation using an inverted automated Nikon TiE fluorescence microscope. The ratio of cells with aggregated foci was calculated by dividing the number of cells with aggregated foci by the total number

of cells. For detecting fluorescently tagged Rnr3, Sml1, Mrc1, Cmr1, and Pol30 by microscopy, cells were inoculated in SC medium containing 100 μg/mL adenine and grown by shaking overnight at 25 °C. Cultures were diluted to $OD_{600} = 0.2$ and grown for one cell cycle prior to microscopy.

Fluorophores were cyan fluorescent protein (clone W7), YFP (clone 10C), and red fluorescent protein (clone yEmRFP or mCherry). Fluorophores were visualized on a Deltavision Elite microscope (Applied Precision, Inc.) equipped with a ×100 objective lens (Olympus U-PLAN S-APO, NA 1.4), a cooled Evolve 512 EMCCD camera (Photometrics, Japan), and an Insight solid-state illumination source (Applied Precision, Inc.). Images were acquired using softWoRx version 7.0.0 (Applied Precision, Inc.) software and fluorescence intensities measured with Volocity software version 5.4 (PerkinElmer).

**Cell cycle progression analyses.** Cells were grown at 25 °C in SC medium supplemented with 2% glucose and synchronized in $G_1$ using 8 μg/ml α-factor (Biotem, France) for 150 min. Cells were released in S-phase from the $G_1$ arrest by α-factor degradation using 75 μg/ml Pronase and 20 mM phosphate buffer. Samples from the time-course experiments were processed with standard methods for subsequent flow cytometry analysis. Data were acquired on a MACSQuant Analyser (Miltenyi Biotec) and analyzed with FlowJo software. Three independent biological replicates have been performed for each cell cycle progression analysis. Cells (10,000) were gated on PI-Height/PI-Area. Haploid and diploid strains were used for calibration. Analysis of flow cytometry profiles was performed in FlowJo version 10. Profiles can be found in Supplementary Fig. 10.

**Protein extraction and Rad53 western blotting.** Approximately $5 \times 10^8$ cells were collected at each relevant time point and were washed with 20% trichloroacetic acid to prevent proteolysis, then resuspended in 200 μl of 20% trichloroacetic acid at room temperature. The same volume of glass beads was added and cells were disrupted by vortexing for 10 min. The resulting extract was spun for 10 min at $1000 \times g$ at room temperature and the resulting pellet resuspended in 200 μl of Laemmli buffer. Whenever the resulting extract was yellow-colored, the minimum necessary volume of 1 M Tris base (non-corrected pH) was added until blue color was restored. Then, water was added until a final volume of 300 μl was reached. These extracts were boiled for 10 min and clarified by centrifugation as before; 10–15 μl of this supernatant was loaded onto a 3–8% acrylamide gradient Invitrogen gel and migrated 70 min at 150 V to separate Rad53 isoforms, then proteins were transferred to a nitrocellulose membrane. Detection by immunoblotting was accomplished with anti-Rad53 antibody, a kind gift from Dr. C. Santocanale, used in a 1/3000 dilution. To quantify total protein, membranes were stained with Ponceau S stain for 5 min. The membrane was destained with several changes of water for 30 s to 1 min each. Afterwards, the blotting was photographed.

**Replication fork speed measurements.** Replication fork progression was quantitatively analyzed following 50 μM 5-ethynyl-2'-deoxyuridine (EdU) incorporation and DNA fiber combing. For more details, see Supplementary Information. EdU replication tracks were detected using Alexa Fluor 555 Azide (ThermoFisher) and Click chemistry. DNA fibers were detected using mouse anti-ssDNA (DSHB, http://dshb.biology.uiowa.edu/autoimmune-ssDNA; autoanti-ssDNA was deposited to the DSHB by Voss, E.W. (DSHB Hybridoma Product autoanti-ssDNA)))[82] and goat anti-mouse coupled to Alexa Fluor 647 (ThermoFisher, A21241). Antibodies were used in a 1/50 dilution. 300 to 600 individual EdU tracks were counted for each experimental condition. Statistical analyses were performed with GraphPad Prism 7.

**Chromatin immunoprecipitation assay.** ChIP experiments were performed essentially as described[60]. Briefly, cells (50 mL per time point) were collected at $OD_{600} = 0.6–0.7$ and were fixed with 1% formaldehyde (Sigma-Aldrich, catalog number 252549) for 10 min at room temperature on a roller table. Next, glycine was added to a final concentration of 120 mM and samples were incubated for an additional 10 min at room temperature and then placed on ice. Cells were collected (3 min, $1000 \times g$, 4 °C), washed in ice-cold HBS (25 mM Hepes pH 8.0, 140 mM NaCl), and frozen in to FastPrep tubes (MP Biomedicals, catalog number 115076200) at −20 °C until further processing. Cell pellets were resuspended in 600 μL cold lysis buffer (25 mM HEPES pH 7.5, 140 mM NaCl, 1 mM EDTA pH 8.0, 1% NP40, 2 mM sodium deoxycholate, 1 mM phenylmethylsulfonyl fluoride, and protease inhibitor cocktail (Roche, catalog number 11836153001)). Glass beads (200 μL, 425–600 μm) were added and cells were disrupted for three cycles of 45 s at maximum power using a Fastprep homogenizer (MP Biomedicals). The glass beads were discarded by filtration and the chromatin collected by centrifugation at $15,000 \times g$ for 30 min at 4 °C. Next, the pellet of cross-linked chromatin was resuspended in 500 μL lysis buffer and sonicated in a pre-cooled waterbath at 4 °C (Misonix Sonicator 3000, 50 cycles of 10 s at setting #3, pause 10 s between each cycle). Finally, 300 μL of fresh lysis buffer was added to each sample before removal of cell debris by centrifugation twice at $7000 \times g$ (5 and 15 min, 4 °C) and transfer to new tubes. After this step, 10 μL of each sample was mixed with 120 μL TE buffer (20 mM Tris-HCl pH 7.5, 1 mM EDTA) containing 1% SDS to serve as INPUT control. Remaining sample volumes were split into three new tubes containing either 1 μL mouse monoclonal anti-Myc antibody (sc-40, 9E10, Santa Cruz

Biotechnology; 1/200 dilution) or 1 μL mouse anti-FLAG, F1804, Sigma-Aldrich; 1/200 dilution) or no antibody, and incubated rotating at 4 °C for 1 h. Next, 20 μL of Protein G Dynabeads (10004D) pre-equilibrated in HBS buffer was added to each tube and incubation continued rotating for 2 h at 4 °C to allow binding of the antibody to the beads. All remaining washes were performed at room temperature. Next, samples were placed on magnet and washed with 1 mL lysis buffer rotating at 4 °C for 5 min. Then, samples were washed rotating in 1 mL AT1 buffer (25 mM HEPES pH 7.5, 140 mM NaCl, 1 mM EDTA, 0.03% SDS (freshly added)) for 5 min. Next, samples were washed rotating in 1 mL AT2 buffer (25 mM HEPES pH 7.5, 1 M NaCl, 1 mM EDTA) for 5 min. Then, samples were washed rotating in 1 mL AT3 buffer (20 mM Tris-HCl pH 7.5, 1 mM EDTA, 250 mM LiCl, 0.5% NP40, 10 mM sodium deoxycholate) for 5 min. Finally, samples were washed twice rotating in 1 mL TE buffer for 5 min. Immunoprecipitated material was eluted from the beads in 155 μL TE buffer containing 1% SDS by heating for 10 min at 65 °C. Beads were removed by placing on a magnet and transferring the supernatant to a new tube. To reverse cross-linking, samples were incubated overnight at 65 °C. After addition of 240 μL TE buffer and 20 μL proteinase K (20 mg/mL stock solution), samples were incubated for 2 h at 37 °C. Next, DNA was extracted by adding 50 μL 5 M LiCl and 450 μL phenol/chloroform, and vortexing for 10 min. Samples were spun for 5 min at $15,000 \times g$ and the water phase was transferred to a new tube, where the DNA was precipitated by addition of 1 mL 96% ethanol, 5 μL glycogen (Roche,catalog number 10901393001) and 50 μL 3 M NaOAc, and incubating overnight at −80 °C. Precipitated DNA was collected by centrifugation for 5 min at $15,000 \times g$, washed with 1 mL 70% ethanol, dried for 30 min at room temperature, and finally resuspended in 50 μL ddH$_2$O. Samples were stored at −20 °C until use.

Real-time quantitative PCR was performed at the early-replicating ARS305, at regions 3 kb upstream and downstream ARS305, and at late-replicating ARS501, using a CFX96 Real-time System (BioRad). The average of six real-time PCR measurements (three biological replicates and two technical replicates for each) and SD is reported. Fold increase of the IP over the beads-only control was calculated[66] using the formula: fold increase = $2^{(C_T\text{input} - C_T\text{IP})}$ per $2^{(C_T\text{input} - C_T\text{beads})}$.

**Statistical analyses.** Statistical significance of differences in mutation rates was assessed using a likelihood ratio test, as implemented in the R package rSalvador. Statistical significance of differences in mutant frequency was assessed using an unpaired $t$-test with Welch's correction. Statistical significance of differences in Mrc1 foci was assessed using a one-tailed Fisher's exact test. Statistical analyses were performed with GraphPad Prism 7, with the exception of statistical analyses of differences in mutation rates.

Additional methodology information can be found in Supplementary Methods.

**Reporting summary.** Further information on research design is available in the Nature Research Reporting Summary linked to this article.

## Data availability
The RNA-sequencing dataset generated and analyzed during the current study has been deposited in Sequence Read Archive, as Bioproject PRJNA632734 (http://www.ncbi.nlm.nih.gov/bioproject/632734). The *CAN1* sequencing data generated and analyzed during the current study has been deposited at GenBank, accession codes MT509124–MT509357. All data are available from the authors upon reasonable request. Source data are provided with this paper.

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

## Acknowledgements

We thank Patricia T.N. Van Dam (TU Delft) for technical assistance with the acetaldehyde HPLC measurements, as well as all CMPG members for discussions, Valmik Vyas for plasmid pV1382 and sharing the protocol for CRISPR, Judith Frydman for sharing the pESC-VHL-mCherry plasmid, and Boris Pfander for sharing plasmid pBP81. This work was financially supported by a grant from Research Foundation - Flanders (FWO)—grant number G090618N. K.V. acknowledges financial support from FWO by a postdoctoral fellowship (1249117 N). L.H. was supported by a PhD fellowship from FWO (11B6720N). J.K. was supported by a KU Leuven F + fellowship. M.L. is supported by The Independent Research Fund Denmark (FNU) and the Villum Foundation. Work in the lab of P.P. is funded by grants from the Agence Nationale pour la Recherche (ANR), the Institut National du Cancer(INCa), and the Ligue contre le Cancer (équipe labellisée). Research in the lab of K.J.V. is supported by KU Leuven Program Financing, European Research Council (ERC) Consolidator Grant CoG682009, Human Frontier Science (HFSP) program grant RGP0050/2013, Vlaams Instituut voor Biotechnologie (VIB), European Molecular Biology Organization (EMBO) Young Investigator program, FWO, and Agentschap voor Innovatie door Wetenschap and Technology (IWT). Research in the P.V.L. lab is supported by the Francis Crick Institute, which receives its core funding from Cancer Research UK (FC001202), the UK Medical Research Council (FC001202), and the Wellcome Trust (FC001202). P.V.L. is a Winton Group Leader in recognition of the Winton Charitable Foundation's support towards the establishment of The Francis Crick Institute.

## Author contributions

K.V., M.L., and K.J.V. conceived the experiments. K.V., L.G., B.P., P.P., M.L., and K.J.V. designed the experiments. K.V., C.C., L.G., K.D., L.H., K.G., P.B., and E.V.d.Z. performed the experiments. K.V., L.H., J.K., P.B., T.S., B.P., J.M., P.V.L., M.L., and K.J.V. contributed analysis ideas. K.V., C.C., L.G., B.P., P.P., M.L., and K.J.V. analyzed the data. K.V. and J.G. analyzed the *CAN1* sequencing data. J.K. analyzed the RNA-sequencing data. K.V., B.P., L.H., J.K., T.S., C.C., J.G., J.M., P.V.L., S.N., P.P., M.L., and K.J.V. wrote the manuscript.

## Competing interests

The authors declare no competing interests.
