## [Peer Review File · Nature Communications]

Reviewers' comments:

Reviewer #1 (Remarks to the Author):

This is a very interesting study that aims to understand the mechanisms by which ethanol acts as a mutagen. Despite the long-established role of ethanol as a carcinogen, its role as a mutagen is still far from clear. In this work it is proposed that ethanol causes replication stress and so an increased mutation rate via the engagement of error-prone polymerases. The genetic experiments performed in yeast are strong and convincing: preventing the recruitment of the error-prone polymerases or inactivating them abolishes the mutagenic effect of ethanol. Recent analyses of cancer genomes have also pinpointed error-prone polymerases as crucial for the mutational impact of ethanol on cancer genomes. The conclusions of this study are therefore very relevant to cancer research.

Minor points:

55-57 ref missing

64 'factors that cause stress' – what kind of stress? Also in later sentences 'stress' should always be specified

all figures. Show the data points instead of bar/dynamite plots. Bar plots hide data distributions.

Fig 1 could be presented more clearly: Label with reporter used (CAN1, URA1), assign letter to each panel (separate a).

195 This paragraph is confusing as the last sentence says that the statements from the previous ones are invalid? If the genetic approach does not alter acetaldehyde levels, it cannot inform about the effect of acetaldehyde. The same is true for the next paragraph.

Fig 3 no error bars in bar charts to show variation between experiments?

Fig 4a The GO terms describing up- and downregulated transcription were the most significant ones or picked by hand?

The logic for studying Mrc1/Claspin is not well explained.

437-453 info in paragraph partly repetitive with what was stated before. Then a new paragraph is started referring to info in the one before? It would be better to merge the two paragraphs in a concise way to avoid repeating information.

Supplementary Figure 6: Why is this important (?) result shifted to the supplement

The connection with cancer is very interesting. However this link to recent work elucidating the role of error prone polymerases in alcohol-associated mutations in tumors only comes in the final lines of the discussion whereas it would be more appropriate to introduce this v relevant work in the introduction. In the discussion it would be good to have a more direct comparison between the yeast and tumour data (e.g. PolZ suggested to be more important in yeast compared to polH in tumors (tho also likely some contribution of both in both systems e.g. a PolZ-like signature suggested in at least one analysis of cancer genomes: <https://www.nature.com/articles/ncomms15290>).

Reviewer #2 (Remarks to the Author):

Description:

The current view is that ethanol metabolism leads to toxic derivatives that form DNA and protein adducts, which are a major cause of ethanol-induced mutagenesis and carcinogenesis. In this study the authors suggest an alternative mechanism, in which ethanol-induced mutations in *S. cerevisiae* are generated due to induction of replication stress by ethanol, which causes slow-down of replication, and generation of single-stranded DNA regions, to which TLS polymerases are recruited and perform error-prone DNA synthesis.

Critique:

This study is of general interest, aiming to gain insight into the mutagenic and carcinogenic activities of ethanol, and proposing a new mechanism of action. However, a key point is that the authors were unable to rule out the involvement of ethanol-induced DNA adducts in the effects that they have observed, most importantly mutagenesis. This is because they report that they did not see any change in acetaldehyde concentration under several manipulations, including knockdown of ADH2 gene (which may be insufficient) or chemical inhibition. In addition, the possibility of the formation of other ethanol derivatives was not examined. This means that the mutations that they observed could have been caused by DNA adducts, which makes the involvement of TLS expected, since it is the major mechanism which generates point mutations at DNA lesions. Also proteotoxic stress can be caused by ethanol-generated protein adducts. All of this means that some key results presented in this manuscript can be attributed to adducts generated by ethanol metabolism, a possibility that the authors do not seriously consider. The arguments described above do not rule out the possibility that ethanol acts also by the mechanism proposed by the authors, perhaps even in combination with adducts effects. However, the authors have not provided convincing evidence to support such a mechanism. Most importantly, they have not shown replication slow-down (they showed inhibition of the cell cycle), and they have not demonstrated the formation of ssDNA stretches in the cells, two key aspects in their model. The authors need more experimental evidence, and perhaps try to integrate the two models for ethanol-induced mutagenesis.

Reviewer #1 (Remarks to the Author):

This is a very interesting study that aims to understand the mechanisms by which ethanol acts as a mutagen. Despite the long-established role of ethanol as a carcinogen, its role as a mutagen is still far from clear. In this work it is proposed that ethanol causes replication stress and so an increased mutation rate via the engagement of error-prone polymerases. The genetic experiments performed in yeast are strong and convincing: preventing the recruitment of the error-prone polymerases or inactivating them abolishes the mutagenic effect of ethanol. Recent analyses of cancer genomes have also pinpointed error-prone polymerases as crucial for the mutational impact of ethanol on cancer genomes. The conclusions of this study are therefore very relevant to cancer research.

We would like to thank the reviewer for his/her kind words. We would also like to thank the reviewer for indicating where the manuscript lacked clarity. Below and in the revised manuscript, we have addressed all the reviewer's comments.

Minor points:

55-57 ref missing

We have now added references for this statement.

64 'factors that cause stress' – what kind of stress? Also in later sentences 'stress' should always be specified

We have now specified the kind of stress. For example, in the introduction lines 66-68 now read: 'What is known, however, is that several stressors, such as nutrient starvation, drug treatment with for example fluconazole and high salinity can affect mutation rates and genome stability across multiple organisms'¹³.

all figures. Show the data points instead of bar/dynamite plots. Bar plots hide data distributions.

*While we agree with the comment that bar plots do not show data distributions, we think that for most the data in our manuscript, bar plots with error bars actually summarize the data and distribution in the most easily accessible way. However, following the reviewer's comment, we have now changed several of our graphs to show the data points – see for example **Figure 5, panels a, b and d**. We have now also included all raw data used to make the figures in the Source Data File associated with the manuscript.*

Specifically, for the mutation rate data (fluctuation assays): mutation rates are obtained using the golden standard approach, namely using the MSS-maximum likelihood method to estimate m (number of mutations) based on results obtained from many parallel cultures (in our case, at least 54 cultures were used per condition!). Ultimately, data from all these cultures is used to estimate mutation rate, and it is this estimation, together with 95% confidence intervals, that is shown in each graph reporting mutation rate. Hence, in this specific case, it is not possible to show the individual data points. This is a standard way to show the results of a fluctuation assay, see for example PMID:23935537; PMID: 30137492.

Fig 1 could be presented more clearly: Label with reporter used (CAN1, URA1), assign letter to each panel (separate a).

We agree with the reviewer. We have now divided the figure into 4 panels, separating panel a into panels a and b in the revised version. We have also added a text label indicating the mutation reporter gene used to each mutation rate graph in this figure. The new figure is appended below:

Figure 1. Ethanol increases mutation rate

a,b. Mutation rate increases with ethanol concentrations.

Cultures of *S. cerevisiae* strain S288c (strain VK111) (a) and RM11-1a (b) were grown in synthetic media (2% glucose) and indicated ethanol concentrations (v/v). For each condition, at least 54 cultures were analyzed. Mutation rates were determined by fluctuation assays on canavanine. For more details, see methods section. Error bars represent 95% confidence intervals. Statistical significance of differences in mutation rates was assessed using a likelihood ratio test. * $P < 0.05$, *** $P < 0.001$.

c. Ethanol does not affect cell viability.

Cells of strain VK111 were grown in synthetic media (2% glucose), at different ethanol concentrations (v/v) for the same time as a standard fluctuation assay. Cell viability was determined using methylene blue staining. Bars represent average of 9 measurements +/- SD. Error bars are clipped at 100%. At least 540 cells were analyzed per ethanol concentration.

d. Effect of ethanol on mutation rate is also observed using URA3 mutation reporter.

Cells of strain VK111 were grown in synthetic media (2% glucose) at different ethanol concentrations (v/v). For each condition, at least 54 cultures were analyzed. Mutation rates were determined by fluctuation assays on 5-fluoro-orotic acid (FOA). Error bars represent 95% confidence intervals. Statistical significance of differences in mutation rates was assessed using a likelihood ratio test. *** $P < 0.001$.

Source data for this figure are provided as a Source Data file.

195- This paragraph is confusing as the last sentence says that the statements from the previous ones are invalid? If the genetic approach does not alter acetaldehyde levels, it cannot inform about the effect of acetaldehyde. The same is true for the next paragraph.

We agree with the reviewer that our original phrasing of this part was confusing. We have now rephrased the relevant sections to make our point more clear. The relevant sections now read (lines 175-190):

*“The *S. cerevisiae* genome encodes 5 alcohol dehydrogenase genes (ADH1-5) involved in ethanol metabolism, with ADH2 encoding the enzyme responsible for converting ethanol to acetaldehyde⁵³. Deleting ADH2 did not affect the mutagenic effect of ethanol, and overexpressing ADH2 did not further increase mutation rate upon ethanol exposure (**Figure 2b**). These results would indicate that acetaldehyde is not required for ethanol to exert its mutagenic effect. However, since acetaldehyde levels did not appear to be altered in these strains, as determined using two different approaches (HPLC and enzymatic assays), we cannot draw any definite conclusions about the contribution of acetaldehyde to ethanol-induced mutagenesis based on these experiments (**Figure 2c and Supplementary Figure 2b**).*

*To get more insight into a potential role of acetaldehyde in the mutagenic effect of ethanol, we next used fomepizole, a well-established alcohol dehydrogenase inhibitor⁵⁴⁻⁵⁶. Similar to altering ADH2 levels, fomepizole addition did not abolish the mutagenic effect of ethanol, but also did not appear to change acetaldehyde levels, again preventing us from drawing any definite conclusions on the contribution of acetaldehyde to the mutagenic effect of ethanol (**Supplementary Figure 2c,d**).”*

Importantly, however, the focus of our study is more on the cascade of events that are triggered in ethanol stress, including slowing down of replication forks and the subsequent recruitment of error-prone polymerases.

Fig 3 no error bars in bar charts to show variation between experiments?

The data shown in this figure represents type of mutations found in the CAN1 ORF of cells exposed to 0 (124 colonies analyzed) or 6% EtOH (116 colonies analyzed). Since the data shown represent the percentage of mutations across all samples in one specific condition, we cannot include error bars. We would like to stress that this is a standard, generally accepted way of showing mutation types identified in reporter genes, see for example the following references: PMID: 23935537 and PMID: 31056389.

*In the revised figure legend (**Figure 3**), we have now included the number of colonies analyzed.*

Fig 4a The GO terms describing up- and downregulated transcription were the most significant ones or picked by hand?

*These are the most significant ones, p-values can be found in **Supplementary Dataset 3**. To make this more clear, we now also refer to this Supplementary Dataset in the legend to **Figure 4a**.*

The logic for studying Mrc1/Claspin is not well explained.

We agree with the reviewer that our initial way of explaining our rationale for studying Mrc1 was not clear. We have now tried to make this more clear, by separating the sections on cell cycle progression and

replication rate from the section that introduces Mrc1. More specifically, the logic for studying Mrc1 is now explained as follows (lines 338-341) :

“Our data show that ethanol slows down replication. Since replication rate is highly dependent on Mrc1, the homolog of metazoan Claspin^{41, 80-82}, an evolutionary conserved component of the replisome that links the replicative helicase with DNA polymerase activities⁸³, we next investigated the effects of ethanol on Mrc1.”

437-453 info in paragraph partly repetitive with what was stated before. Then a new paragraph is started referring to info in the one before? It would be better to merge the two paragraphs in a concise way to avoid repeating information.

We agree with the reviewer. We have now rewritten both the section on Mrc1 as well as the starting paragraph on the involvement of error-prone polymerases to avoid repetition (lines 338-370 in revised manuscript). More specifically, the starting paragraph on the involvement of error-prone polymerases now reads as follows:

“Replication forks lacking Mrc1 progress more slowly and have been reported to lead to DNA damage^{41, 80, 82, 86}. Translesion polymerases are recruited to sites of replication fork stalling and/or DNA damage. Interestingly, several reports demonstrate that these polymerases are recruited to defective or stalled replisomes, without a need for actual DNA damage to occur^{37, 87}. Translesion polymerases have a higher error-rate than the regular replicative polymerases and are hence sometimes referred to as error-prone polymerases^{22, 88}.”

Supplementary Figure 6: Why is this important (?) result shifted to the supplement.

*We have now made this figure a main figure – **Figure 8**.*

The connection with cancer is very interesting. However this link to recent work elucidating the role of error prone polymerases in alcohol-associated mutations in tumors only comes in the final lines of the discussion whereas it would be more appropriate to introduce this v relevant work in the introduction.

We have now mentioned the link between error-prone polymerases and mutations in alcohol-associated tumors in the introduction (lines 80-83):

“Interestingly, an elegant study recently showed that alcohol-associated cancers display error-prone polymerase-associated mutational spectra, although the exact mechanism by which these polymerases are involved and/or are affected by ethanol remained unclear²⁴.”

We also elaborate this in more detail in the discussion (see also next comment).

In the discussion it would be good to have a more direct comparison between the yeast and tumour data (e.g. PolZ suggested to be more important in yeast compared to polH in tumors (tho also likely some contribution of both in both systems e.g. a PolZ-like signature suggested in at least one analysis of cancer genomes: <https://www.nature.com/articles/ncomms15290>).

We thank the reviewer for this suggestion, and we have now elaborated on this in the discussion. More specifically, the section in the discussion is as follows (lines 496-501):

“Our data links error-prone polymerases to alcohol-related mutations in S. cerevisiae. Interestingly, a recent study reported the presence of error-prone polymerase-associated mutational spectra in alcohol-related tumors²⁴. It should be noted that in the latter case, tumor samples displayed a mutational spectrum characteristic for PolH (encoded by RAD30 in S. cerevisiae), whereas our data implicate PolZ as the primary source of alcohol-related mutations in S. cerevisiae.”

Reviewer #2 (Remarks to the Author):

We would like to thank this reviewer for the constructive criticism. The reviewer raised valid points regarding the interpretation and discussion of our results. We have now addressed all points in full. We have added experimental data on (i) replication rate and (ii) ssDNA formation in ethanol-exposed cells. We have also added a more elaborate discussion of the potential mechanism(s) involved.

Description:

The current view is that ethanol metabolism leads to toxic derivatives that form DNA and protein adducts, which are a major cause of ethanol-induced mutagenesis and carcinogenesis. In this study the authors suggest an alternative mechanism, in which ethanol-induced mutations in *S. cerevisiae* are generated due to induction of replication stress by ethanol, which causes slow-down of replication, and generation of single-stranded DNA regions, to which TLS polymerases are recruited and perform error-prone DNA synthesis.

Critique:

This study is of general interest, aiming to gain insight into the mutagenic and carcinogenic activities of ethanol, and proposing a new mechanism of action. However, a key point is that the authors were unable to rule out the involvement of ethanol-induced DNA adducts in the effects that they have observed, most importantly mutagenesis. This is because they report that they did not see any change in acetaldehyde concentration under several manipulations, including knockdown of ADH2 gene (which may be insufficient) or chemical inhibition.

We are happy to read that this reviewer agrees that our study is of general interest. We completely agree with the reviewer that we have not shown with 100 % certainty that acetaldehyde, and acetaldehyde-derived adducts, are not involved in the mutagenic effect of ethanol.

*For the sake of completeness, we would like to point out that, apart from the experiments listed in the manuscript, we have tried multiple other experimental approaches to investigate the potential involvement of acetaldehyde and acetaldehyde-derived adducts. As the results listed below illustrate, acetaldehyde levels seem to be extremely difficult to manipulate and/or measure correctly (at least in *S. cerevisiae*).*

More specifically, we have tried the following approaches to investigate the importance of acetaldehyde in mediating the mutagenic effect of ethanol:

- **Chemical approaches**

- **Extracellular addition of acetaldehyde – included in manuscript**

*As we show in **Figure 2a** and **Supplementary Figure 2** of our paper, addition of extracellular acetaldehyde either killed cells (toxicity) or, in lower concentrations, did not increase mutation rate.*

- **Disulfiram – not included in manuscript**
Disulfiram inhibits acetaldehyde dehydrogenase, and addition of this drug should lead to increased acetaldehyde levels since it blocks the conversion of acetaldehyde to acetate. Although we observed that disulfiram increases mutation rate, we could not detect increased acetaldehyde levels in disulfiram-treated cells.
- **Fomepizole – included in manuscript**
*Fomepizole is a well-established alcohol dehydrogenase inhibitor⁵³⁻⁵⁵. Fomepizole addition did not abolish the mutagenic effect of ethanol (see **Supplementary Figure 2**), but also did not appear to change acetaldehyde levels, again preventing us from drawing any definite conclusions on the role of acetaldehyde in the mutagenic effect of ethanol.*
- **Genetic manipulation**
The yeast genome encodes 5 alcohol hydrogenases involved in ethanol metabolism, ADH1 to ADH5. Of these, Adh1 and Adh3-5 reduce acetaldehyde to ethanol during glucose fermentation; whereas Adh2 is the only enzyme that is reported to be responsible for oxidizing ethanol to acetaldehyde.
 - **ADH2 – included in manuscript**
*ADH2 encodes the enzyme responsible for converting ethanol to acetaldehyde⁵². Deleting ADH2 did not affect the mutagenic effect of ethanol, and overexpressing ADH2 did not further increase mutation rate upon ethanol exposure (**Figure 2b**). These results would indicate that acetaldehyde is not required for ethanol to exert its mutagenic effect. However, since acetaldehyde levels did not appear to be altered in these strains, as determined using two different approaches (HPLC and enzymatic assays – see also below), we cannot draw any definite conclusions about the role of acetaldehyde based on these experiments.*
 - **ADH1 – not included in manuscript**
We next tested the effect of altering ADH1 levels on the mutagenic effect of ethanol. Since Adh1 converts acetaldehyde into ethanol, overexpressing ADH1 should lead to decreased acetaldehyde levels and, if acetaldehyde is mediating the mutagenic effect of ethanol, consequently result in a reduced mutation rate. However, we find that overexpressing ADH1 does not reduce the mutagenic effect of ethanol.
 - **Other ADH genes – not included in manuscript**
Although ADH1 and ADH2 encode the main enzymes affecting acetaldehyde levels, it could be possible that some of the other ADH genes are masking the effect of altering ADH1/ADH2 levels. To try to investigate this possibility, we used different quadruple deletion strains, in which all ADH genes, except for one, were deleted (insert reference). Unfortunately (and perhaps not surprisingly), all these strains were very sick, showing extremely slow growth rates on a multitude of carbon sources. This prevented us from accurately determining mutation rates.

Additionally, we have also used two different methods to determine acetaldehyde levels (HPLC and enzymatic assays); but neither of these two methods could detect a difference in acetaldehyde levels in our conditions tested (**results included in manuscript**).

In this regard, it is interesting to note that we have also used the HPLC method to determine acetaldehyde levels in *adh* mutants of *E. coli*. We find that in this case, deleting the two main acetaldehyde metabolizing enzymes does reduce acetaldehyde levels (**panel b** in figure below), while mutation rate still increases when these mutant cells are exposed to ethanol (**panel c** in figure below). In other words, at least in *E. coli*, acetaldehyde does not seem to be mediating the mutagenic effect of ethanol (**results not included in manuscript**).

Conversion of ethanol to acetaldehyde plays no role in ethanol- induced mutagenesis in Escherichia coli. **a**, The role of AdhE and AdhP in the interconversion between ethanol and acetaldehyde. **b**, The intracellular acetaldehyde levels increase upon exposure to 3% ethanol in the wild type. In the double *adh* knock-out however, no significant increase in acetaldehyde levels was observed (mean s.d., repeated measures one-way ANOVA with post-hoc Tukey correction; ns: not significant; ****: $P < 0.0001$). **c**, The mutation rate in the double *adh* mutant increases with increasing ethanol concentrations, demonstrating that the conversion of ethanol to acetaldehyde plays no role in the ethanol- induced mutation rate.

Taken together, these data illustrate how difficult it appears to be to alter and/or measure acetaldehyde levels in yeast cells – something we also point out in our manuscript.

We have deliberately not included all data shown above in our revised manuscript, since none of these experiments offer any definitive conclusions about the role of acetaldehyde in the mutagenic effect of

ethanol in yeast cells. We have made this also more clear in the revised version of the manuscript, see also our reply to reviewer 1's comments.

While our manuscript was under revision, a new study showed that, in vitro, acetaldehyde causes GG to TT mutations, due to the formation of GG intrastrand crosslinks. Earlier analysis of a reporter gene in human cells indicated that a similar reaction also takes place in vivo. We did not observe any GG to TT transitions in the CAN1 ORF of EtOH-treated cells. This further supports the notion that, in our conditions, acetaldehyde is not the main mediator of the mutagenic effect of ethanol. We have added these analyses to our revised manuscript (lines 244-249).

Most importantly, we want to point out that the main focus of our study is on what happens downstream of the DNA damage and/or replication fork stalling – where we show that replication forks stall and that this is followed by recruitment of error-prone polymerases, which eventually lead to mutations.

In addition, the possibility of the formation of other ethanol derivatives was not examined. This means that the mutations that they observed could have been caused by DNA adducts, which makes the involvement of TLS expected, since it is the major mechanism which generates point mutations at DNA lesions.

We fully agree with the reviewer that the involvement of other ethanol derivatives is an important issue. The carcinogenic effects of ethanol have also been linked to reactive oxygen species (ROS) produced during ethanol metabolism. ROS can cause lipid peroxidation and the subsequent formation of (mutagenic) DNA adducts. To investigate if ROS could mediate the observed mutagenic effect of ethanol in our conditions, we assessed ROS production by measuring oxidation of H₂DCFDA (2',7'-dichlorofluorescein diacetate), a commonly used oxidant sensitive probe, to a fluorescent product. As expected, we find that hydrogen peroxide causes oxidation of H₂DCFDA. Ethanol-exposed cells on the other hand do not display an increase in fluorescence. These results indicate that ethanol exposure does not cause an increase in ROS. Hence, it seems unlikely that ROS are involved in the mutagenic effect of ethanol. We have added this data to our revised manuscript (lines 217-228):

*“The carcinogenic effects of ethanol have also been linked to reactive oxygen species (ROS) produced during ethanol metabolism¹¹. ROS can cause lipid peroxidation and the subsequent formation of (mutagenic) DNA adducts. We assessed ROS production in EtOH-treated cells using the cell permeant reagent H₂DCFDA (2',7'-dichlorofluorescein diacetate), a commonly used oxidant sensitive probe^{64, 65}. After diffusion into the cell, H₂DCFDA is first deacetylated by cellular esterases. In the presence of ROS, this probe is then readily oxidized into a fluorescent compound. As expected, we find that exposing cells to hydrogen peroxide increased oxidant levels ($p=0.0029$, unpaired t-test with Welch's correction) (**Supplementary Figure 4**). Ethanol exposed cells on the other hand do not cause an increase in fluorescence. These results indicate that ethanol exposure does not cause an increase in ROS.”*

Supplementary Figure 4. Measurement of ethanol-induced oxidation.

Cells (VK111) were grown in synthetic medium (2% glucose) and incubated with the oxidant-sensitive probe H₂DCFDA and the indicated chemicals, after which fluorescence was analysed using flow cytometry. Three replicates of 50 000 cells were analysed per condition. Bars represent average +/- SD. Statistical significance was assessed using an unpaired t-test with Welch's correction. **P < 0.01.

Also proteotoxic stress can be caused by ethanol-generated protein adducts.

We completely agree with the reviewer that ethanol-generated protein adducts could cause proteotoxic stress. Multiple studies have identified ethanol-induced protein adducts. In fact, the reported adducts have been mainly attributed to ethanol metabolism. More specifically, acetaldehyde and ROS molecules produced during ethanol metabolism can react with proteins to form adducts. While our data seems to indicate that acetaldehyde and ROS are not involved in the mutagenic effect of ethanol (see also previous comments), we cannot rule out the presence of such adducts in our set-up. Alternatively, ethanol can cause proteotoxic stress through protein misfolding or unfolding. We have now included a part on the potential role of protein adducts in the discussion of our revised manuscript (lines 433-441):

“One interesting hypothesis is that the proteotoxic stress observed in ethanol-exposed cells could be due to ethanol-generated protein adducts. In fact, multiple studies have identified various ethanol-induced protein adducts ^{11, 100}. These adducts have been mainly attributed to ethanol metabolism. More specifically, acetaldehyde and ROS produced during ethanol metabolism can react with proteins and form adducts. While our data seems to indicate that ROS and likely also acetaldehyde are not responsible for the mutagenic effect of ethanol, it is possible that ethanol-generated adducts, perhaps together with other sources of ethanol-derived proteotoxic stress, such as denatured proteins, could underlie the observed mutagenic effect of ethanol; potentially by affecting replication fork components.”

We have also modified **Figure 9** to better reflect this –see below.

All of this means that some key results presented in this manuscript can be attributed to adducts generated by ethanol metabolism, a possibility that the authors do not seriously consider.

We agree with the reviewer that our results indeed do not completely rule out the (additional?) involvement of some products of ethanol metabolism in the mutagenic effect of ethanol. While we already briefly mentioned this in the original version of our manuscript, we have now both added more experimental data (see ROS data mentioned above) as well as discussed the potential involvement of

adducts in more detail in the discussion (lines 420-432; plus 433-441 on ethanol-generated protein adducts). We have also updated **Figure 9** to reflect this.

The relevant section now reads:

“Furthermore, our experiments are not conclusive about the potential role of acetaldehyde, since it proved difficult to manipulate and measure acetaldehyde levels. However, it is clear that the mechanisms underlying the mutagenic effect of ethanol are more complex than previously thought. Ethanol and/or acetaldehyde could cause chemical damage to the DNA, which causes replication fork stalling and recruitment of error-prone translesion polymerases. However, it seems equally plausible that the recruitment of these error-prone polymerases is directly caused by the proteotoxic effect of ethanol on the replication fork, causing it to become unstable and stall. In fact, both mechanisms are not mutually exclusive and difficult to disentangle. In the case of lesions due to chemical DNA damage (for example caused by acetaldehyde-derived adducts), we would expect replication fork collapse and a strong checkpoint activation. Interestingly, we do not observe a strong checkpoint activation by ethanol, indicating that replication forks do not collapse; again pointing to the complex mechanisms underlying the mutagenic effects of ethanol.”

The arguments described above do not rule out the possibility that ethanol acts also by the mechanism proposed by the authors, perhaps even in combination with adducts effects. However, the authors have not provided convincing evidence to support such a mechanism.

We have now performed the following additional experiments to further investigate the mechanisms underlying the mutagenic effect of ethanol

- *Involvement of ROS in mutagenic effect of ethanol – **Supplementary Figure 4** in revised manuscript – already discussed above.*
- *Investigate checkpoint activation by ethanol – **Supplementary Figure 7** in revised manuscript*

Lines 317-323: “we find that Rad53, the effector kinase that is phosphorylated after activation of either the DNA damage checkpoint or the DNA replication checkpoint, is not phosphorylated in

response to ethanol treatment (**Supplementary Figure 7**). We also can not detect a significant increase in ssDNA levels in ethanol-exposed cells (**Supplementary Figure 6**), with ssDNA accumulation being a signal for checkpoint activation. Together, this data indicate that ethanol only causes a mild replication stress checkpoint activation, one that is much less pronounced compared to MMS.”

- Investigate replication rate – **Figure 5** in revised manuscript – see detailed reply below.
- Investigate formation of ssDNA stretches – **Supplementary Figure 6** in revised manuscript –see detailed reply below.

Taken together, our data suggest a model whereby EtOH causes both proteotoxic and replication stress, and where ethanol exposure results in Mrc1 relocalization from the replication fork to INQ, resulting in a less stable and slower replication fork. This in turn triggers exchange of the regular DNA polymerase for error-prone polymerases, which ultimately leads to increased mutation rates.

Based on this data and the reviewers’ suggestions, we have also significantly reworked the discussion to better reflect the complex mechanism(s) underlying the mutagenic effect of ethanol.

Most importantly, they have not shown replication slow-down (they showed inhibition of the cell cycle)

We have now determined replication fork progression by pulsed incorporation of EdU and DNA combing of DNA fibers isolated from cells exposed to 0 and 6% EtOH (this is the standard way to check for replication slow-down, see for example PMID:18353973, PMID:16137625; PMID:16631586). Analysis of distribution of EdU tracts in both conditions shows that ethanol exposure significantly reduces EdU track length (see new **Figure 5**; median track length of 27.2 kb in 0% EtOH, compared to median track length of 19.7 kb in 6% EtOH; P-value < 0.0001; Mann-Whitney unpaired non-parametric t-test). This data indicate that the cell cycle delay observed in ethanol-exposed cells is caused by altered DNA replication and suggest that ethanol affects replication fork progression.

Figure 5. Ethanol affects replication rate.

a. Ethanol weakly activates the replication checkpoint.

Haploid cells expressing YFP-Sml1 (IG101-12D) were exposed to 0 or 6% EtOH for 2h and imaged using fluorescence microscopy. Ethanol-exposed cells display decreased Sml1 levels. Data represent average fluorescence intensities of individual cells., Error bars represent 95% confidence intervals. At least 198 cells were analyzed per condition. Statistical significance was assessed using an unpaired t-test with Welch's correction. *** $P < 0.001$. AU, arbitrary units.

b. Ethanol increases cellular Rnr3 protein levels.

Haploid cells expressing YFP-Rnr3 (W6986-1B) were exposed to 0 or 6% EtOH for 2h or 0.03% MMS for 1h and imaged using fluorescence microscopy. Data represents average fluorescence intensity of individual cells. Error bars represent 95% confidence intervals. At least 291 cells were analyzed per condition. Data represents average fluorescence intensity, error bars represent 95% confidence intervals. Statistical significance was assessed using an unpaired t-test with Welch's correction. *** $P < 0.001$.AU, arbitrary units.

c. Cell-cycle progression is slower in ethanol-exposed cells.

Wild-type cells were arrested in G_1 with α -factor and were released synchronously into S phase with the addition of pronase, in medium containing 0 or 6% EtOH. Cells were collected at the indicated time points. DNA content was assessed using flow cytometry.

The 1C peak corresponds to cells in the G0/G1 phase. The 2C peak (double the amount of fluorescence intensity thus double the amount of DNA) corresponds to cells in the G2/M phase.

d,e. Replication fork progression is slowed down by ethanol.

Fork speed, measured after pulse incorporation of EdU and DNA combing, was analyzed in asynchronous cell cultures (strain PP2226) exposed for 2 hours to 0 or 6% ethanol. For each condition, at least 339 cells coming from 3 independent replicates were analyzed. The scatter dot plot depicts the distribution of EdU track lengths. Medians are shown by a red line and are indicated. Statistical significance was assessed using a Mann-Whitney unpaired non-parametric t-test. **** P<0.0001. **e** Examples of the DNA fibers (green) containing EdU tracks (red) in each condition. EdU tracks are also highlighted in white below each fiber.

Source data for this figure are provided as a Source Data file.

, and they have not demonstrated the formation of ssDNA stretches in the cells, two key aspects in their model.

We have now checked for the formation of ssDNA stretches in cells upon exposure to ethanol. We have done this by using the presence of Rfa1-YFP foci as a measure for ssDNA, with Rfa1 a ssDNA-binding protein. We find that cells not exposed to EtOH display mostly a diffuse nuclear localization of Rfa1, consistent with no ssDNA being present (**panel a** of figure below). When cells go through S phase in the presence of MMS (a DNA alkylating agent known to cause ssDNA), cells progressively accumulate Rfa1 foci/speckles, reflecting the collapse of replication forks at MMS-induced DNA lesions (**panel c** of figure below). Interestingly, when cells go through S phase in the presence of 6% ethanol, we do not observe an increase of Rfa1 foci/speckles significantly different from the untreated cells (**panel b** of figure below), indicating that the replication fork remains intact in ethanol, but perhaps moves more slowly, consistent with our fiber analysis. This slow-moving replication fork can trigger recruitment of error-prone polymerases. We have added this data as **Supplementary Figure 6**.

Supplementary Figure 6. Single-stranded DNA does not form extensively upon ethanol exposure. Wild-type (ML147-2B) cells were arrested in G1 by alpha-factor and released into S phase in the absence or presence of 6% ethanol or 0.03% MMS. Cells were analyzed for DNA content (right panels) and the presence of Rfa1-YFP speckles/foci as a measure for ssDNA at the indicated time points (left panels). Representative microscopy images are shown above each column in the graph. Scale bars, 3 μ m. Error bars indicate 95% confidence intervals. For each time point, two biological replicates with a total of 187-847 cells were analyzed. Y-axis truncated at the value 10 for better display.

Reviewers' comments:

Reviewer #1 (Remarks to the Author):

The authors have addressed my concerns and have improved the manuscript. I really enjoyed this thought-provoking study.

Reviewer #2 (Remarks to the Author):

The authors have made an effort to address the concerns raised. However, I am still puzzled by the lack of any involvement of acetaldehyde derived adducts in the mutagenesis. Particularly puzzling is the lack of effect on mutation frequency upon acetaldehyde treatment, contrary to earlier published results. If the lack of effect of acetaldehyde is not a technical problem, it may be supporting the author's argument of additional mechanisms for ethanol-induced mutagenesis, because it would mean that at least in the strain they studied, ethanol, but not acetaldehyde, caused mutations.

To sort this out, I suggest the following:

1. The effect of acetaldehyde on mutations should be tested in 2 other yeast strains. If an effect of acetaldehyde is found, it can serve as a 'positive control'. It may also indicate that the effect they originally observed is strain specific, but it may still be generally relevant.
2. If at least one of the strains in item 1 shows increased mutations, the combined action of ethanol plus acetaldehyde should be tested. Are they additive? The two may react to form an acetal, but it would be important to test it in any case.
3. The authors argue that they did not find in the CAN1 ORF a GG-to-TT mutation, typical of acetaldehyde-induced mutations. They should check whether this type of tandem double mutation, if formed, can generate a canavanine-resistant mutation. It might not be possible
4. Experiments 1 and 2 should be repeated also with another mutagenesis assay system (URA3? Other?)

If this point is sorted out, the manuscript will be acceptable.

Reviewer #1 (Remarks to the Author):

The authors have addressed my concerns and have improved the manuscript. I really enjoyed this thought-provoking study.

We would like to thank the reviewers for his/her kind words.

Reviewer #2 (Remarks to the Author):

The authors have made an effort to address the concerns raised. However, I am still puzzled by the lack of any involvement of acetaldehyde derived adducts in the mutagenesis. Particularly puzzling is the lack of effect on mutation frequency upon acetaldehyde treatment, contrary to earlier published results. If the lack of effect of acetaldehyde is not a technical problem, it may be supporting the author's argument of additional mechanisms for ethanol-induced mutagenesis, because it would mean that at least in the strain they studied, ethanol, but not acetaldehyde, caused mutations.

To sort this out, I suggest the following:

1. The effect of acetaldehyde on mutations should be tested in 2 other yeast strains. If an effect of acetaldehyde is found, it can serve as a 'positive control'. It may also indicate that the effect they originally observed is strain specific, but it may still be generally relevant.

To address comment 1 and comment 4 of this reviewer, we carried out additional fluctuation assays using 5 genetically different yeast strains, 4 different acetaldehyde concentrations and 2 different mutation reporters (URA3 and CAN1).

Importantly, to be able to use these different mutation reporters accurately, yeast strains need to be haploid and have an intact functional copy of either URA3 (for detection of mutants using FOA) or CAN1 (for detection of mutants using canavanine). These requirements prevented us from testing all strains in all conditions.

Results of these experiments are in Figure below.

CAN1 mutation reporter

URA3 mutation reporter

Supplementary Figure 4. Effect of acetaldehyde on mutation rate in different yeast strains.

Cultures of different yeast strains were grown in synthetic media (2% glucose) and indicated acetaldehyde concentrations (v/v). For each condition, at least 54 cultures were analyzed. Mutation rates were determined by fluctuation assays on canavanine or FOA. Error bars represent 95% confidence intervals. Statistical significance of differences in mutation rates was assessed using a likelihood ratio test. * $P < 0.05$, *** $P < 0.001$.

We find that, similar to the prototrophic S288c strain we have used throughout our manuscript, acetaldehyde also does not increase mutation rate in an auxotrophic S288c strain (panel a and b, CAN1 as mutation reporter).

The two other yeast strains we used with the CAN1 mutation reporter are RM11-1a and W303. The sequence divergence between RM11-1a is around 0.5-1% (comparable to the sequence divergence between human and chimp). This sequence variation is distributed throughout the genome, confirming that RM11 shares no recent history with S288c. Around 85% of the genome sequence of W303 is derived from S288c. In total, around 800 ORFs differ between W303 and S288c. For RM11-1a and W303, we observe that acetaldehyde causes a slight increase in mutation rate (panels c and d – please note that W303 did not grow at an acetaldehyde concentration of 0.1%, while other strains grew much slower compared to non-treated cultures, suggesting that this concentration is really at the borderline of what yeasts can tolerate).

As for the strain tested using URA3 as a mutation reporter, we see that acetaldehyde strongly increases mutation rate in strain YJM789 (panel e). The YJM789 genome sequence is marked by extensive polymorphisms relative to S288c throughout the nuclear and mitochondrial genomes. The ≈60,000 SNPs scattered over the genome alignment represent a SNP frequency of 1 in 164 bp (0.6%).

Interestingly, exposing cells to acetaldehyde generally results in a lower fold increase in mutation rate compared to when cells are exposed to ethanol. For example, ethanol causes an up to 3.5 fold increase in mutation rate in a prototrophic S288c strain, whereas acetaldehyde results in a 1.2 fold increase in this strain. For RM11-1a, the difference is even more pronounced: ethanol causes a 3.9 fold increase in mutation rate, whereas acetaldehyde results in a 1.2 fold increase.

Taken together, these results indicate that the lack of mutagenic effect of acetaldehyde in S288c is not due to a technical problem. Instead, our results show that in some yeast strains, acetaldehyde is mutagenic, as was also previously reported. We agree with reviewer 2 that this data further supports our manuscript's argument for additional mechanisms for ethanol-induced mutagenesis. We also think that the differences between strains are interesting and merit further research, possibly in follow-up studies.

We have added this figure as Supplementary Figure 4 to our revised manuscript. The corresponding section (lines 212-221 of revised manuscript) now reads:

“To further investigate the effect of acetaldehyde exposure on mutation rate, we carried out additional fluctuation assays using a total of 5 genetically different yeast strains, 4 different acetaldehyde concentrations and 2 different mutation reporters (URA3 and CAN1). Our results show that in some, but not all of the yeast strains tested, acetaldehyde increases mutation rate (Supplementary Figure 4). We also noticed that the highest concentration of acetaldehyde that we used in our assays (0.1%) was borderline lethal, suggesting that the acetaldehyde clearly affected the cells' functioning. These results indicate that the lack of mutagenic effect of acetaldehyde observed in some strains is not due to a technical problem, and provides additional support that the mutagenic effect of ethanol observed in yeast is not solely due to acetaldehyde.”

2. If at least one of the strains in item 1 shows increased mutations, the combined action of ethanol plus acetaldehyde should be tested. Are they additive? The two may react to form an acetal, but it would be important to test it in any case.

We agree that this would be an interesting follow-up experiment, but we also feel that this really is outside of the direct scope of this paper. The main point of our paper is the mutagenic effect of ethanol and the involvement and recruitment of error-prone polymerases in this process.

3. The authors argue that they did not find in the CAN1 ORF a GG-to-TT mutation, typical of acetaldehyde-induced mutations. They should check whether this type of tandem double mutation, if formed, can generate a canavanine-resistant mutation. It might not be possible.

The CAN1 ORF contains 180 GG pairs, and some of these can generate stop codons and non-synonymous mutations when mutated to TT. Hence, it seems plausible that some GG-to-TT mutations would generate canavanine-resistant colonies. The fact that we did not observe this type of mutations in our ethanol-treated cells seems to indicate that the observed mutations in ethanol-exposed cells are not induced by acetaldehyde. We have added this information to our revised manuscript (lines 256-259). The corresponding section in our manuscript now reads:

“Interestingly, both in vivo and in vitro studies have indicated that acetaldehyde causes GG to TT mutations, due to the formation of GG intrastrand crosslinks^{59, 66}. The CAN1 ORF contains 180 GG pairs, and some of these can generate stop codons and non-synonymous mutations when mutated to TT. Hence, it seems plausible that some GG-to-TT mutations would generate canavanine-resistant colonies. We did not observe any GG to TT mutations in the CAN1 ORF isolated from ethanol-treated cells. In line with our previous observations (Figure 2 and Supplementary Figure 2-3), this could again indicate that, in our conditions, acetaldehyde is not the main mediator of ethanol-induced mutagenesis.”

4. Experiments 1 and 2 should be repeated also with another mutagenesis assay system (URA3? Other?)

As outlined in our reply to comment 2 of this reviewer, we have now tested the effect of acetaldehyde using the two most commonly used mutagenesis assay systems in yeast, namely URA3 (selection of mutants on FOA) and CAN1 (selection of mutants on canavanine).

If this point is sorted out, the manuscript will be acceptable.

REVIEWERS' COMMENTS:

Reviewer #2 (Remarks to the Author):

The authors have satisfactorily responded to my criticism, and I recommend that the manuscript is accepted for publication.

I congratulate the authors on this important contribution to the mechanistic insight on the mutagenesis and potentially carcinogenesis caused by ethanol.

Zvi Livneh

Answers to referee comments

REVIEWERS' COMMENTS:

Reviewer #2 (Remarks to the Author):

The authors have satisfactory responded to my criticism, and I recommend that the manuscript is accepted for publication.

I congratulate the authors on this important contribution to the mechanistic insight on the mutagenesis and potentially carcinogenesis caused by ethanol.

Zvi Livneh

We want to thank this reviewer for his thorough assessment of the paper; and his comments that helped to further improve the manuscript.